# Circulating MiR-30b-5p is upregulated in Cavalier King Charles Spaniels affected by early myxomatous mitral valve disease

Mara Bagardi[1], Sara Ghilardi[1], Valentina Zamarian[2], Fabrizio Ceciliani[1], Paola G. Brambilla[1]*, Cristina Lecchi[1]

1 Department of Veterinary Medicine and Animal Science, University of Milan, Lodi, Italy, 2 Diabetes Research Institute, IRCCS San Raffaele Hospital, Milan, Italy

* paola.brambilla@unimi.it

**Data Availability Statement:** All relevant data are within the paper and its Supporting Information files.

## Abstract

There is a growing interest in developing new molecular markers of heart disease in young dogs affected by myxomatous mitral valve disease. The study aimed to measure 3 circulating microRNAs and their application as potential biomarkers in the plasma of Cavalier King Charles Spaniels with early asymptomatic myxomatous mitral valve disease. The hypothesis is that healthy Cavalier King Charles Spaniels have different microRNA expression profiles than affected dogs in American College of Veterinary Internal Medicine (ACVIM) stage B1. The profiles can differ within the same class among subjects of different ages. This is a prospective cross-sectional study. Thirty-three Cavalier King Charles Spaniels in ACVIM stage B1 were divided into three groups (11 younger than 3 years, 11 older than 3 years and younger than 7 years, and 11 older than 7 years), and 11 healthy (ACVIM stage A) dogs of the same breed were included as the control group. Three circulating microRNAs (miR-1-3p, miR30b-5p, and miR-128-3p) were measured by quantitative real-time PCR using TaqMan® probes. Diagnostic performance was evaluated by calculating the area under the receiver operating curve (AUC). MiR-30b-5p was significantly higher in ACVIM B1 dogs than in ACVIM A subjects, and the area under the receiver operating curve was 0.79. According to the age of dogs, the amount of miR-30b-5p was statistically significantly higher in group B1<3y (2.3 folds, $P = 0.034$), B1 3-7y (2.2 folds, $P = 0.028$), and B1>7y (2.7 folds, $P = 0.018$) than in group A. The area under the receiver operating curves were fair in discriminating between group B1<3y and group A (AUC 0.780), between B1 3-7y and A (AUC 0.78), and good in discriminating between group B1>7y and A (AUC 0.822). Identifying dogs with early asymptomatic myxomatous mitral valve disease through the evaluation of miR-30b-5p represents an intriguing possibility that certainly merits further research. Studies enrolling a larger number of dogs with preclinical stages of myxomatous mitral valve disease are needed to expand further and validate conclusively the preliminary findings from this report.

**Funding:** The authors received no specific funding for this work.

**Competing interests:** The authors have declared that no competing interests exist.

## Introduction

Myxomatous mitral valve disease (MMVD) is a cardiovascular disease affecting dogs, progressing from mitral regurgitation to heart failure [1]. Although MMVD seems to be a genetic disorder, veterinary studies on molecular genetics have not provided conclusive data on causative mutations in MMVD development and progression [2–4]. The incidence is age-related and is exceptionally high in some breeds, such as the Cavalier King Charles Spaniel (CKCS). Half of the CKCSs are estimated to be affected by MMVD at the age of 6–7, while at 10 years of age, almost all of them are [2, 5–7]. Evidence from highly susceptible breeds such as CKCS and Dachshund shows a robust inherited component to the disease and suggests a polygenic inheritance [3, 8, 9]. Due to the lack of early clinical signs and predictive biomarkers, early diagnosis is difficult. Transthoracic echocardiography is currently the gold standard for diagnosing MMVD [10]. However, this test needs specialized equipment and well-trained operators to reduce inter-observer variability, as valves affected by mild changes may be misinterpreted as normal.

For this reason, identifying reliable specific biomarkers is desirable, especially for screening and breeding programs. In human medicine, microRNAs (miRNAs) are potentially suitable markers of cardiovascular diseases [11, 12]. MiRNAs exert their function by repressing the translation of target genes and regulating protein production through different mechanisms in several pathophysiological conditions, including myocardial infarction, hypertrophy, fibrosis, and inflammation. MiRNAs can be secreted into extracellular fluids, including plasma and serum, within vesicles, such as exosomes, or with lipoproteins and RNA-binding proteins, namely Argonaute. They are relatively stable even under challenging conditions such as long-time storage at room temperature and low or high pH [13–17]. Aberrant expression of miRNAs is associated with several human [18–21] and veterinary [22–25] disorders, including cancer and heart diseases. With specific regard to canine MMVD, the dysregulation of circulating miRNAs has been previously investigated by different approaches, including real-time quantitative PCR, microarray, and next-generation sequencing [26–31]. However, it should be noticed that most of the dogs enrolled in these studies were classified as American College of Veterinary Internal Medicine (ACVIM) stages C and D, while only one previous study has investigated circulating miRNAs in adult dogs with preclinical MMVD (i.e., ACVIM stages B1 and B2) [29–32].

It should be also highlighted that data from previous researches on dysregulation of circulating miRNAs in dogs with MMVD are partially biased by the limited sample sizes and the heterogeneity of the therapeutic protocols as well as study populations, as they were not specifically focused exclusively on one specific stage of the disease [28, 29, 31, 33, 34]. Given the above, this study was aimed at investigating the potential use of miRNA as biomarkers to identify dogs belonging to a specific ACVIM class, namely the stage B1 [32]. The decision to focus on this ACVIM class was driven by the fact that these dogs are most subjected to breed screening, and therefore are targeted as potential breeders. Moreover, the inclusion of a single ACVIM class offered the advantage of investigating miRNA in a very homogenous study population, which, in this case, is also free from the potential confounding effects of cardiovascular drugs (as ACVIM stage B1 do not need medical treatment for their underlying structural heart disease) [32].

To further standardize our study population we also selected a specific canine breed, namely the CKCS, being the most commonly affected by MMVD [2, 3, 5–9]. Lastly, we focused our analysis on three specific miRNAs previously demonstrated to be associated with canine MMVD [29, 31, 33] and arbitrarily divided our study population in three distinct age categories (see below for further details) to explore how these miRNAs are modulated in the plasma of CKCS of different ages affected by MMVD at stage ACVIM B1.

## Materials and methods

### Clinical and echocardiographic examinations

The study included owned CKCSs visited at the Cardiology Unit of the Department of Veterinary Medicine, University of Milan, between May 2019 and July 2020. According to the University of Milan's ethical committee statement, informed consent was signed by the owners, number 2/2016, and a high standard of care was provided throughout each examination.

A cardiological evaluation was performed on dogs that fasted for at least 12 hours during a routine veterinary visit. The clinical data of the animals included animal history and clinical and echocardiographic examinations. The cardiovascular system was evaluated by checking the presence/absence of murmurs by two operators with different levels of expertise, respectively a third-year PhD student in cardiology and a professor with more than twenty years of practice in clinical veterinary cardiology (MB and PGB). The evaluated auscultatory findings included presence/absence, timing, and intensity of the murmur (0 = absent; 1 = I-II/VI left apical systolic or soft; 2 = III-IV/VI bilateral systolic or moderate and loud respectively; 3 = V-VI/VI bilateral systolic or palpable) [35]. Blood pressure was indirectly measured with a Doppler method according to the ACVIM consensus statement [36]. Peripheral venous blood sampling was performed at the end of the examination. Blood was collected from the jugular or cephalic vein in two 2.5-mL EDTA tubes. Part of the blood samples was used to perform routine bloodwork (i.e., complete blood cell count and serum biochemistry) to rule out possible comorbidities.

The echocardiographic exam was used to diagnose MMVD. A standard transthoracic echocardiographic examination was performed with My Lab50 Gold Cardiovascular ultrasound machine (Esaote, Genova, Italy), equipped with multi-frequency phased array probes (3.5–5 and 7.5–10 MHz), chosen according to the weight of the subject. Videoclips were acquired and stored using the echo machine software for offline measurements. The exam was performed by a third-year PhD student (MB) according to a standard procedure with concurrent continuous electrocardiographic monitoring [37]. All examinations were performed without pharmacological restraint. Dogs were classified according to the ACVIM classification scheme [32].

Inclusion criteria for dogs in the clinically normal group (ACVIM A) were: no echocardiographic evidence of heart disease, no clinical signs, no abnormalities on results of a complete blood count and biochemical analyses, and no history of medical treatment within the previous 6 months. Inclusion criteria for dogs with MMVD at stage B1 were: no abnormalities on results of a complete blood count and biochemical analyses, echocardiographic evidence of a thickened or prolapsed mitral valve and mitral valve regurgitation, no evidence of left atrial dilatation, defined as a left atrial-to-aortic root ratio (LA/Ao) <1.6 on 2-dimensional echocardiography, and no left ventricle dilation, defined as left ventricular end-diastolic diameter normalized for body weight (LVIDdN) <1.7. Dogs that presented left atrial and/or ventricular remodeling, but were not severe enough to meet the current guidelines criteria for ACVIM class B2, were also included [32]. The degree of mitral regurgitation (jet size) was assessed using the color Doppler by calculating the maximal ratio of the regurgitant jet area signal to the left atrium area [38]. The regurgitant jet size was estimated as the percentage of the left atrial area (to the nearest 5%) occupied by the larger jet. It was considered as trivial or trace (<10%), mild (between 10 and 30%), moderate (between 30 and 70%) or severe (>70%) [38, 39]. More in detail, mitral regurgitation was considered trivial when the regurgitant jet was not detectable in all systolic events, while it was considered a trace when it was always visible [39]. Furthermore, dogs in ACVIM class B1 had no medical treatment history within the previous 6 months. Four groups of 11 client-owned dogs were included in the present study: group A, or healthy control, group B1<3 with dogs younger than 3 years; group B1 3–7, with dogs older than 3 years and younger than 7 years, and B1>7 with dogs older than 7 years [40, 41].

Dogs with asymptomatic MMVD and cardiac remodeling (ACVIM stages B2), dogs with symptomatic MMVD (ACVIM stages C and D), or with other systemic diseases such as systemic hypertension, uncontrolled hypothyroidism, hyperadrenocorticism, primary pulmonary hypertension, neoplasia, and other cardiac abnormalities such as dilated cardiomyopathy, congenital cardiac abnormalities, endocarditis, and severe arrhythmia, or that had received any drugs in the last 6 months, were excluded from the study. The type of diet was not an exclusion criterion for this study.

## Small RNA isolation and real-time quantitative PCR quantification

Blood samples for small RNA isolation were collected in 2.5 mL EDTA-K3 tubes. Within 2 hours, the samples were centrifuged at 800 × g for 15 minutes. Plasma was stored at –80˚C until RNA isolation.

Small RNA was extracted using the miRNeasy Serum/Plasma Kit (Qiagen, catalogue number 217184, Milan, Italy). An aliquot of 150 μL of plasma per sample was thawed on ice and centrifuged at 3000 × g for 5 minutes at 4˚C. RNA was extracted using miRNeasy Serum/Plasma Kits (Qiagen, catalogue number 217184, Milano, Italy) following the manufacturer's instructions. One mL of Qiazol (Qiagen) was added to an aliquot of 150 μL of plasma per sample. After incubation at room temperature for 5 minutes, 25 fmol of the exogenous synthetic spike-in control *Caenorhabditis elegans* miRNA cel-miR-39 (Qiagen, catalogue number 219610) was spiked into samples at the beginning of the extraction procedure to check both the extraction of miRNAs and the efficiency of the complementary DNA (cDNA) synthesis. RNA extraction was then carried out according to the manufacturer's instructions. RNA yield and successful RNA purification without contamination of proteins or residues from the isolation procedure were assessed using 1 μL of eluted RNA applied to a NanoDrop ND-1000 spectrophotometer. The 260/280 nm ratio was between 1.8 and 2.2 for all RNA samples, and the range of 260/230 nm ratio was from 2.0 to 2.2 according to MIQE guidelines [42, 43]. To obtain cDNA, reverse transcription was performed on 10 ng of total RNA using a TaqMan Advanced miRNA cDNA Synthesis Kit (catalogue number A28007, Applied Biosystems) following the manufacturer's instructions.

Real-time quantitative- PCR was performed following the MIQE guidelines [42, 43]. The small RNA TaqMan assays were performed according to the manufacturer's instructions using the selected primer/probe assays (ThermoFisher Scientific), which are also specific for canine miRNAs, including: cel-miR-39-3p (assay ID 478293_mir); miR-1-3p (assay ID 477820_mir) [31]; miR-30b-5p (assay ID 478007_mir) [33]; miR-128-3p (assay ID mmu480912_mir) [31]. The reference miRNA was miR-16-5p (assay ID rno481312_mir). Quantitation was performed on 15 μL in a CFX Connect Real-Time PCR Detection System (Bio-Rad) using 7.5 μL of 2X TaqMan Fast Advanced Master Mix (Cat. No. 4444557), 0.75 μL of miRNA specific TaqMan Advanced assay reagent (20X), 1 μL of cDNA, and water to make up the remaining volume. The thermal cycling profile was as follows: 50˚C for 2 minutes, 95˚C for 3 minutes, 40 cycles at 95˚C for 15 s and 60˚C for 40 s. No-RT controls and no-template controls were included. MicroRNA expressions are presented in fold change normalized to miR-16 as reference miRNA and sample A as reference sample using the formula $2^{-\Delta\Delta Cq}$ on Bio-Rad CFX Maestro Software [43].

## Statistical analysis

Statistical analysis was performed using XLStat software for Windows (Addinsoft, New York, USA). Data were tested for normality using the Shapiro–Wilk test; the nonparametric Kruskal-Wallis test was applied when the data were not normally distributed. Receiver

operating characteristic (ROC) analysis was performed, and the area under the ROC curve was considered a measure of the diagnostic accuracy using the definition suggested by Šimundić in 2009 [44]. The diagnostic value was calculated for miRNA that showed significant differential expression in the canine blood. Statistical significance was accepted at a $P$-value $\leq 0.05$, and all the significance values were adjusted according to the Bonferroni post-hoc correction.

## Results

### Demographics and characteristics of study subjects

The median age of the 44 included CKCSs was 3.3 years (IQR$_{25\text{-}75}$ 1.81–6.99), and the median body weight was 8.1 Kg (IQR$_{25\text{-}75}$ 7.48–9.68). Fourteen subjects (31.82%) were males, and 30 (68.18%) were females. Study population characteristics (clinical and echocardiographic data), grouped according to the ACVIM classes and, for the B1 class, to the age of MMVD diagnosis, are shown in Table 1. Weight was lower in B1<3 ($P = 0.040$) and A ($P = 0.029$) subjects compared with the B1>7 group, whereas echocardiographic variables were not statistically different among groups of age ($P > 0.05$).

**Table 1. Clinical and echocardiographic data of all included CKCSs divided according to the ACVIM classification scheme and age at the time of the MMVD diagnosis for subjects belonging to ACVIM stage B1.**

| | Overall population | A | B1<3 | B1 3–7 | B1>7 |
|---|---|---|---|---|---|
| **Number of dogs** | 44 | 11 | 11 | 11 | 11 |
| **Sex** | 30F (8NF) | 8F (1NF) | 6F | 6F (1 NF) | 10F (6 NF) |
| | 14M | 3M | 5M | 5M | 1M |
| **Age (years)** | 3.30 (1.81–6.99) | 1.96 (1.73–2.88) | 1.52 (1.07–2.21) | 3.88 (3.49–4.30) | 8.14 (7.66–8.68) |
| **Weight (kg)** | 8.10 (7.48–9.68) | 7.75 (7.25–7.95) [a] | 7.80 (6.83–8.00) [a] | 9.40 (7.63–9.88) | 10.00 (9.35–10.40) |
| **SBP (mmHg)** | 135 (110–145) | 125 (115–135) | 130 (120–140) | 130 (120–140) | 140 (125–150) |
| **Murmur** | 29 grade 0 | | | | |
| | 10 grade 1 | | 10 grade 0 | 8 grade 0 | 6 grade 1 |
| | 5 grade 2 | 11 grade 0 | 1 grade 1 | 3 grade 1 | 5 grade 2 |
| **Regurgitant jet size** | 11 grade 0 | | | | |
| | 5 grade 1 | | | | |
| | 8 grade 2 | | | | |
| | 15 grade 3 | | 5 grade 1 | 2 grade 2 | 6 grade 3 |
| | 5 grade 4 | 11 grade 0 | 6 grade 2 | 9 grade 3 | 5 grade 4 |
| **LA/Ao** | 1.15 (1.08–1.26) | 1.21 (1.09–1.36) | 1.08 (1.02–1.20) | 1.17 (1.12–1.19) | 1.15 (1.08–1.29) |
| **E (m/s)** | 0.73 (0.66–0.80) | 0.76 (0.71–0.84) | 0.68 (0.61–0.79) | 0.70 (0.67–0.75) | 0.68 (0.67–0.81) |
| **E/A** | 1.30 (1.18–1.43) | 1.29 (1.21–1.37) | 1.47 (1.36–1.60) | 1.21 (1.16–1.45) | 1.22 (0.99–1.34) |
| **EF (%)** | 67 (58–73) | 67 (62–77) | 62 (59–70) | 68 (57–74) | 67 (57–71) |
| **FS (%)** | 35 (30–40) | 35 (32–43) | 31 (30–38) | 36 (29–41) | 37 (28–39) |
| **LVIDSN** | 0.84 (0.77–0.95) | 0.82 (0.70–0.84) | 0.82 (0.78–0.90) | 0.87 (0.79–0.96) | 0.90 (0.81–0.98) |
| **LVIDDN** | 1.36 (1-29-1.51) | 1.31 (1.26–1.42) | 1.33 (1.22–1.43) | 1.37 (1.33–1.62) | 1.48 (1.40–1.55) |

E = E wave velocity; E/A = E and A waves ratio; EF = ejection fraction (M-mode measurement); FS = shortening fraction (M-mode measurement); LA/Ao = left atrium to aortic root ratio; LVIDDN = normalized left ventricular internal diameter in diastole; LVIDSN = normalized left ventricular internal diameter in systole;

Murmur = left systolic heart murmur intensity: 0 = absent, 1 = I-II/VI left apical systolic, or soft, 2 = III-IV/VI bilateral systolic, or moderate and loud; Sex: F = female, NF = neutered female, M = male; SBP = systemic blood pressure; Regurgitant jet size: 0 = absent, 1 = trivial, 2 = trace, 3 = mild, 4 = moderate.

All data are expressed as median and IQR25-75 ranges.

Values within a row differ significantly at P < 0.05 from B1>7 subjects.

## MiR-30b-5p is differentially expressed in myxomatous mitral valve disease-affected dogs

Small RNA was extracted from plasma, and the spike-in cel-miR-39 was quantified in all collected samples exhibiting a mean Cq of 26.09 (SD 1.13). Three miRNAs, namely miR-1-3p, miR-30b-5p, and miR-128-3p, were detected in all plasma samples (Fig 1A–1F). The comparative analysis demonstrated that one miRNA, namely miR-30b-5p, had a significant differential amount in the plasma of MMVD-affected dogs compared to the healthy group. In detail, the amount of miR-30b-5p increased 2.4 folds ($P = 0.006$) in group B1 compared to group A (Fig 1B). When group B1 was further split according to the age of dogs, the expression of miR-30b-5p remained significantly higher (Fig 1E): group B1<3 (2.3 folds $P = 0.034$), B1 3–7 (2.2 folds $P = 0.028$), and B1>7 (2.7 folds $P = 0.018$) showed a higher level of miR-30b-5p than group A. No differences were found in the amount of miR-1-3p (Fig 1A and 1D) and miR-128-3p (Fig 1C and 1F). The age proved not to be correlated with the expression of the analyzed miRNAs, neither in the entire population nor in each age class ($P > 0.05$).

## Diagnostic performance of MiR-30b-5p discriminating between myxomatous mitral valve disease-affected and healthy dogs

ROC curve analysis was performed to evaluate the diagnostic value of miR-30b-5p in plasma, and the associated AUC was used to confirm the diagnostic potency. Cut-off points were set to maximize the sum of sensitivity and specificity. The ability of miR-30b-5p to separate the tested samples into healthy (stage A) or MMVD-affected (stage B1) was defined as "diagnostic accuracy". It was measured by the area under the curve (AUC). MiR-30b-5p proved to be efficient in discriminating between groups A and B1 (AUC = 0.79; 95% CI 0.65–0.93) (Fig 2A). Even after dividing group B1 according to age, it could efficiently discriminate between group A and group B1<3 (AUC = 0.78; 95% CI 0.60–0.96) and group A and group B1 3–7 (AUC = 0.78; 95% CI 0.60–0.96) (Fig 2B and 2C, respectively), but in particular it proved to be

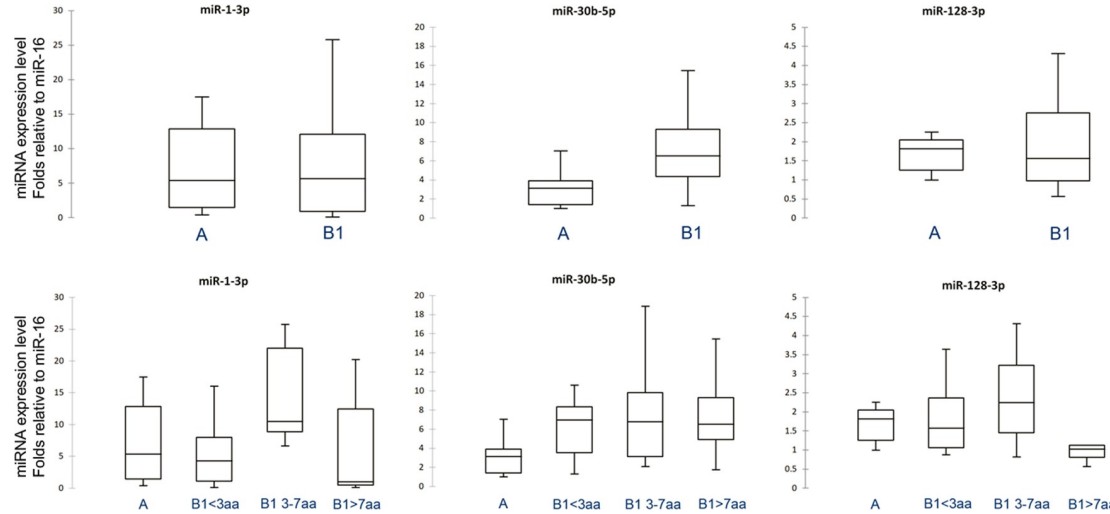

**Fig 1. Expression levels of miR-1-3p, miR-30b-5p, and miR-128-3p between groups.** Expression levels of the three miRNAs between groups A and B1 (A-C, respectively) and between A and B1 were divided according to the age at MMVD diagnosis (D-F, respectively). MiR-30b-5p increased 2.4 folds ($P < 0.05$) in group B1 compared to A (B). Splitting group B1 according to the age of dogs, the expression of miR-30b-5p remained significantly higher (E). Group B1<3 (2.3 folds, $P = 0.034$), B1 3–7 (2.2 folds, $P = 0.028$), and B1>7 (2.7 folds, $P = 0.018$) expressed a higher level of miR-30b-5p than group A. No differences were found in the amount of miR-1-3p (A and D) and miR-128-3p (C and F).

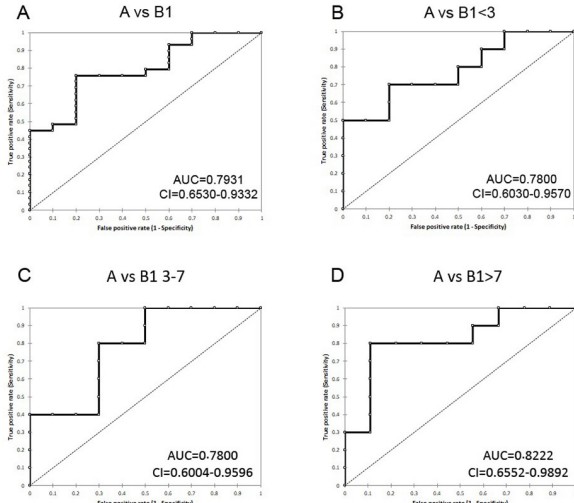

**Fig 2. ROC curves for miR-30b-5p.** Discrimination capacity between group A and group B1 (A), group A and group B1<3 (B), group A and group B1 3–7 (C), and group A and B1>7 (D). MiR-30b-5p can discriminate between healthy and asymptomatic MMVD-affected dogs (stage B1).

really effective in discriminating group A from B1>7 (AUC = 0.82; 95% CI 0.65–0.99) (Fig 2D) (Table 2). Thus, miR-30b-5p can discriminate between healthy (stage A) and asymptomatic MMVD-affected dogs (stage B1).

## Discussion

The present study reported the relationship between the amount of circulating miR-30b-5p and the presence of MMVD, a disease often associated with congestive heart failure (CHF), even in young CKCSs. Our results showed that miR-30b-5p is significantly upregulated in asymptomatic MMVD-affected CKCSs (ACVIM stage B1) than in healthy (ACVIM stage A) dogs. We further demonstrated that miR-30b-5p upregulation is also detectable in young dogs (age <3, ranging from 6 months to 2.4 years), even in MMVD-affected subjects without audible heart murmurs.

Yang and colleagues investigated the cargo of exosomes purified from the plasma of MMVD-affected dogs using an array-based approach, demonstrating that when the False Discovery Rate was set at 20%, 78 miRNAs were dysregulated. Compared to a False Discovery Rate of 10%, no differences were pointed out in either the exosome miRNAs or the whole plasma [28]. Another study, including old dogs (ranging from 8.2 to 13.8 years) with CHF secondary to MMVD (ACVIM stage C), reported that 326 miRNAs were differently modulated comparing healthy (ACVIM stage A) to CHF-affected dogs (ACVIM stage C); the validation

**Table 2. The AUC (95% confidence interval), cut off values, and sensitivity and specificity of miR-30b-5p in CKCSs' plasma.**

|  | AUC | 95% CI | *P*-value | Cut off | Se-Sp |
|---|---|---|---|---|---|
| **A vs B1** | 0.793 | 0.653–0.933 | > 0.0001 | 3.98 | 0.759–0.800 |
| **A vs B1<3** | 0.780 | 0.603–0.957 | 0.0019 | 4.52 | 0.800–0.700 |
| **A vs B1 3–7** | 0.780 | 0.600–0.959 | 0.0023 | 4.37 | 0.800–0.700 |
| **A vs B1>7** | 0.822 | 0.655–0.989 | 0.0002 | 4.37 | 0.800–0.889 |

AUC = area under the ROC curve; CI = confidence interval; Se = sensitivity; Sp = specificity.

step, performed by real-time quantitative PCR, demonstrated the upregulation of miR-133, miR-1, let-7e, and miR-125, and the downregulation of miR-30c, miR-128, miR-142, and miR-423 [31]. Although the study focused on a group of animals affected by a severe disease with clinically detectable signs, the results appeared worthy to be further investigated even in younger patients, prompting us to include miR-1 and miR-128 in the present study. Based on other results previously reported [33] of a study that included old dogs (range, 10.17 ± 3.36 years), we identified miR-30b-5p as a potential marker for further investigation in a younger cohort of MMVD-affected CKCSs of ACVIM stage B1.

Since the molecular background of MMVD is not fully elucidated yet, identifying any specific markers (prognostic and/or therapeutic) would be of great clinical value for recognizing asymptomatic patients, especially at a young age. The diagnosis of MMVD is based on the echocardiographic evaluation of the mitral valve and its leaflets' thickness, which sometimes is hard to quantify since mildly affected valves work adequately, and the lesions don't affect hemodynamics, given the absence of cardiac remodeling and clinical signs. Myxomatous mitral valve disease is age-related, and the prevalence in old small-breed dogs is up to 100%, particularly in chondrodystrophic breeds such as Cocker Spaniels, Dachshunds, and Beagles. Cavalier King Charles Spaniels are more susceptible to developing CHF due to MMVD at a younger age than other breeds [1]. Thus, especially in susceptible breeds such as the CKCS, MMVD occurs at a very young age and progresses over time in different and unpredictable ways. Myxomatous mitral valve disease development is a hereditary character in this breed and has been associated with a multi-factorial polygenic transmission mode. Therefore, several genes are involved, and a defined expression threshold must be reached before the disease occurs [2, 3, 9].

Although miRNAs are intensively investigated in human medicine because of their diagnostic potential in many different conditions, only a few reports are related to circulating miRNAs studies in dogs affected by MMVD. There is no study about the early diagnosis of this disease in a predisposed breed such as the CKCS.

This study identified a biomarker that may impact the implementation of breeding programs through genetic selection and clinical practice. These results confirm that in CKCSs, as already demonstrated in humans, there is a differential expression of miRNAs, suggesting that their expression profiles are distinct for dogs with MMVD compared to healthy dogs. We demonstrated that miR-30b-5p could discriminate among ACVIM stage A CKCSs and ACVIM stage B1 CKCSs younger than 3 years, without audible heart murmurs, but with an echocardiographic diagnosis of MMVD. Our results disagree with the previously reported findings on Dachshunds [33], which demonstrated that miR-30b decreased in the plasma of ACVIM stage B subjects compared with ACVIM stage A. This contradictory result could be explained firstly by the different enrolment strategies (the previous study considered all ACVIM B dogs without distinguishing B1 from B2), and secondly by the different ages of enrolled patients; the present investigation focused mainly on young dogs (33 out of 44 dogs were younger than 4 years), while the previous study only considered old dogs (range, 10.17 ± 3.36 years) [33].

Furthermore, the miR-30 family is abundantly expressed in the heart, and its decrease is strictly related to several heart diseases that result in ventricular remodeling [45]. The dogs enrolled in our study did not have ventricular remodeling yet, but only slightly affected valves. Since the miR-30 family exerts antiapoptotic and anti-inflammatory activities [46, 47], we hypothesized that the expression of miR-30b may increase during the early stage of MMVD in young CKCSs to protect the cardiomyocytes from inflammation and apoptosis and oppose to the atrioventricular valves remodeling.

Identifying dogs with early asymptomatic MMVD through the evaluation of miR-30b-5p is interesting as new clinical research data on this breed and the affected canine population. The possible clinical use of the miR-30b-5p for screening and breeding programs in the CKCS needs more robust data, which could help clinicians and breeders focus on screening programs better and select the breeders carefully. Patients with these characteristics should then be subjected to a closer follow-up. For these reasons, miRNAs may be candidates as novel biomarkers and may provide the basis for further investigations to assess the follow-up and characterize the evolution of the disease in the CKCS [48] without evading the echocardiographic evaluation, which undoubtfully remains the gold standard for MMVD diagnosis [10].

Our results may pave the way towards incorporating this new generation of biomarkers in the traditional diagnostic approach, currently based simply on physical and echocardiographic examinations, to achieve a prompter and more accurate identification of affected dogs [10]. This, in turn, may contribute to treat the disease appropriately from its early stages and, hopefully, decrease the mortality rate of MMVD in CKCSs. If confirmed, this last result would be revolutionary as it would radically change the prognostic perspectives of the most widespread canine heart disease and the most common cause of death in many small-sized canine breeds [1, 10, 49].

Findings from the present research may open a florid collaboration with biomedical companies to develop rapid in-clinic/at-home diagnostic devices to rapidly, accurately, and efficiently evaluate the expression of selected miRNAs. This would further strengthen the technical collaboration between clinicians and biotechnologists (leading to a reciprocal scientific enrichment) and would have a potentially positive economic impact on involved biomedical companies. Indeed, it should be note that, to date, a rapid, not expensive, widely available diagnostic kit to measure the miRNAs amount is not available in small animal practice; therefore, such an analysis is currently limited to few highly specialized laboratories.

This study presents some limitations. The utility of circulating miRNAs as biomarkers of many diseases has attracted considerable attention. However, it is also worth pointing out that the clinical application of miRNAs as biomarkers is still limited. One of the most significant obstacles is the difficulty of normalization circulating miRNAs. Spiked in synthetic miRNAs are widely used to normalize serum and plasma miRNAs expression, but this approach does not include the effects of pre-analytic variables on circulating miRNAs measurement [50, 51].

On the other hand, endogenous miRNAs might be considered suitable reference miRNAs since their expression is affected by the same variables as the targeted miRNAs. A universally accepted normalization strategy is still lacking. The two main strategies involve the identification of a stably expressed reference miRNA previously reported in the literature, or if a miRNA profile has been performed by micro-array or sequencing technologies, the calculation of the global mean expression value of all expressed miRNAs in a given sample [52]. Thus, the selection of different normalization strategies may affect miRNAs quantification and create divergences between studies. We used miR-16 as a reference based on previously reported data, being aware that this is one of the many methods that could be used [33, 53]. The difficulties associated with hemolysis and platelet contamination of plasma samples are also significant. Still, it is conceivable that this issue can be mitigated by reducing the degree of red blood cell and platelet-derived miRNAs contamination with adequate centrifugation and plasma collection [54]. Furthermore, in human medicine it has been demonstrated a potential link between patient's age and miRNAs profile. For example, it has been showed that miRNAs may play a role in the highly coordinated mechanisms regulating genes involved in children's development [55]. Regrettably, in veterinary medicine, to date, such a link has not been investigated. For this reason, veterinary studies purposefully designed to take into account subjects' age categories and investigate the link between age and miRNAs profile are needed. Other limitations

of this study include the small sample size, which should be implemented, and the need for a larger validation group.

## Conclusions

The growing interest in developing new molecular markers of heart disease in young dogs affected by MMVD has led to study the expression of 3 circulating microRNAs and their application as potential biomarkers in the plasma of CKCS with early asymptomatic MMVD (ACVIM stage B1). The hypothesis that healthy dogs have different microRNA expression profiles than B1 subjects of the same breed and that the microRNA profiles can differ within the same class among subjects of different ages have been confirmed. The amount of miR-30b-5p is significantly higher in ACVIM B1 CKCS than in ACVIM A subjects, and according to the age, the amount of miR-30b-5p is higher in group B1 younger than 3 years, B1 between 3 and 7 years, and B1 older than 7 years than in group A. These results lay the basis for future studies aimed at reaching more substantial data that will help the CKCS breeders in their targeted selection programs based on the echocardiographic evaluation and to obtain healthier subjects with a reasonable life expectancy. At the same time, highlighting the risk of developing the disease at an earlier stage will favour a focused screening of the subjects. To that end, identifying early biomarkers for premature MMVD would be a helpful addition.

## Supporting information

**S1 File.**
(XLSX)

## Acknowledgments

The authors are grateful to the many dog owners and breeders for their enthusiastic participation in this work and their availability.

## Author Contributions

**Conceptualization:** Cristina Lecchi.

**Data curation:** Cristina Lecchi.

**Formal analysis:** Valentina Zamarian, Cristina Lecchi.

**Investigation:** Mara Bagardi, Valentina Zamarian, Paola G. Brambilla.

**Methodology:** Mara Bagardi.

**Project administration:** Mara Bagardi.

**Supervision:** Fabrizio Ceciliani, Paola G. Brambilla.

**Writing – original draft:** Mara Bagardi.

**Writing – review & editing:** Sara Ghilardi, Valentina Zamarian, Paola G. Brambilla, Cristina Lecchi.

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
