## [Decision Letter · Decision Letter 0]

18 Apr 2022

PONE-D-22-07425Circulating miR-30b-5p is up regulated in Cavalier King Charles Spaniels affected by early myxomatous mitral valve diseasePLOS ONE

Dear Dr. Brambilla,

Thank you for submitting your manuscript to PLOS ONE. After careful consideration, we feel that it has merit but does not fully meet PLOS ONE’s publication criteria as it currently stands. Therefore, we invite you to submit a revised version of the manuscript that addresses the points raised during the review process.

ACADEMIC EDITOR: All issues raised by expert reviewers are required.

We look forward to receiving your revised manuscript.

Kind regards,

Vincenzo Lionetti, M.D., PhD

Academic Editor

PLOS ONE

Journal Requirements:

 [NO]. 

Reviewers' comments:

Reviewer's Responses to Questions

**Comments to the Author**

1. Is the manuscript technically sound, and do the data support the conclusions?

Reviewer #1: Partly

Reviewer #2: Yes

2. Has the statistical analysis been performed appropriately and rigorously? 

Reviewer #1: Yes

Reviewer #2: Yes

3. Have the authors made all data underlying the findings in their manuscript fully available?

Reviewer #1: Yes

Reviewer #2: Yes

4. Is the manuscript presented in an intelligible fashion and written in standard English?

Reviewer #1: Yes

Reviewer #2: Yes

5. Review Comments to the Author

Reviewer #1: The present study reports interesting clinical research data on a emerging topic in veterinary cardiology (i.e. circulating microRNAs in spontaneus cardiovascular diseases). The study is well conducted and the manuscript is well written. Please consider the comments below.

Abstract: Please revised the abstract based on the comments below.

M&M

Line 104 – “MB” is here identified as “certified cardiologist” and as “third year PhD student” at lines 90-91. Please uniform, or clarify the training of the operator (“diplomate, resident, PhD student”?).

Lines 114-115 – Please correct the term “left ventricular normalized dimensions in diastole (LVIDdN)“ in to “left ventricular end diastolic diameter normalized for body weight (LVIDdN)” as used in the 2019 ACVIM guidelines. And specify if M-mode or 2D was used for this measurement, and which view was used.

Results

Table 1 – Data regarding ESVI and EDVI are reported, but these variables are not described in the M&M. Were ESVI and EDVI obtained using the M-mode (Teichholz formula) and the 2D (e.g. Simpson method)? Specify. In the first case (M-mode derived volumes), I would consider not to include in the study because normalized linear measurements (LVIDDN and LVISDN) are sufficiently informative and most commonly used nowadays in veterinary cardiology.

Table 1 – Please standardize the abbreviations “LVIDas” and “LVIDad” to most used terms: “(LVIDDN) normalized left ventricular internal diameter in diastole” and “(LVIDSN) normalized left ventricular internal diameter in systole”.

Table 1 – “EF%”, please specify if it is M-mode or 2D derived in the M&M.

Table 2 – Please indicate the unit of measurement of the cut-off in the table.

I would recommend to also analyze if B1 dogs without a cardiac murmur have different circulating levels of miR-30b-5p in comparison to B1 dogs with a murmur (independently for age); describing a possible cutoff to predict the presence of a murmur.

Similarly, it could be interesting to describe a possible cutoff of miR-30b-5p in discriminating dogs with a regurgitant jet size of 0-1-2 grade versus 3-4 grade. Please consider this implementation of the study.

Lines 304-306 – The authors state: “The identification of dogs with early asymptomatic MMVD through the evaluation of miR-30b-5p could help the clinicians and the breeders to better focalized screening programs in this breed and to better select the breeders”. This sentence seems to suggest that, based on the present study, the use of miR-30b-5p should be consider for MMVD screening and breeding programs in CKCSs. Personally, I believe that the clinical application of the study results is relatively questionable at this time, because of the small sample size and the not excellent sensitivity of the test (i.e. around 80%). Please remember that screening test are expected to have the highest sensitivity as possible. Echocardiography remains the gold standard for screening and breeding programs (Pedersen HD, et al. Echocardiographic mitral valve prolapse in cavalier King Charles spaniels: epidemiology and prognostic significance for regurgitation. Vet Rec 1999. Birkegård AC, et al. Breeding Restrictions Decrease the Prevalence of Myxomatous Mitral Valve Disease in Cavalier King Charles Spaniels over an 8- to 10-Year Period. J Vet Intern Med. 2016). The results of the present study are interesting as new clinical research data on the disease. The possible clinical use of the miR-30b-5p for screening and breeding programs in CKCS needs stronger data.

Discussion

Line 243 – Please verify the term “abundance”. “circulating levels” or “levels” maybe better?

Line 284 – Please verify the term “preventive”. “Screening” or “breeding” programs mays be better?

Conclusions

The authors state “The present results lay the basis for a breeding program that will help the CKCS’ breeders in their targeted selection to obtain healthier subjects with a reasonable life expectancy”. Again, this conclusion is too strong in my personal opinion and can be a misleading message for the reader. Echocardiography is the non-invasive gold standard method for MMVD diagnosis, and the diagnostic accuracy of miR-30b-5p levels are good, but not enough to consider it a screening test. Considering the common breeding age of 1-4 years, a sensitivity of around 80% and a specificity of around 70% (described in the results) are relatively low for thinking using miR-30b-5p levels in the screening of subjects possibly used for reproduction. There is a relatively high risk of including affected dogs in the program. Personally, I suggest to reconsider the conclusion message.

Similarly, the authors state “…will favour a preventive screening and a mitigating therapeutical approach”. Again, I feel this conclusion as too strong based on the results of the study. Especially I would avoid any reference to “therapeutical” considerations.

Reviewer #2: The manuscript describes the use of a novel type of biomarker, namely microRNA, in dogs affected by myxomatous mitral valve disease. The study’s idea as well as the topic of research are innovative and brilliant. Similarly, a great work has been done by the Authors to combine clinical and cardiological expertise and techniques with laboratory ones. The findings described herein are interesting and the data reported may lay the fundaments for further studies on this topic. Given the above, it was a great opportunity for me to Review this manuscript, and I congratulate with Author for their great job. Below some comments, questions and suggestions aimed at expanding further the results of the study and provide additional information to readers.

Abstract

-lines 22-23: “The aim of the study was to measure the abundance of 3 circulating microRNAs…” Probably, it could be deleted the word “abundance”, as it indirectly implies a result. At the beginning of a study on new field of veterinary medicine, it is almost impossible to be sure that a specific biomarker would be for sure “abundant” or “scant” in a specific population. Therefore, I think it would be more appropriate for the introduction of the abstract this type of sentence: “The aim of the study was to measure 3 circulating microRNAs…”; then, in the result section, it could be stated that “abundance” of the biomarker has been found in a part of the study population.

(*What has been written here, however, should not be considered by the Author if the term “abundance” is the one specifically proposed by the experts in the field of miRNA analysis, and if it is used by them not like an adjective but like a technical word).

-line 25: “33 dogs affected…” Not sure it is allowed to start a sentence with a number. Please, check rules of the Journal and eventually correct here and later in the text.

-line 29: “This is a prospective cross-sectional study”. The sentence that declares the type of study design typically is put before the description of the details of the study population. Accordingly, I would put it before the sentence starting with “33 dogs affected…”.

-lines 29-31: “The abundance of three circulating microRNAs (miR-1-3p, miR30b-5p, and miR-128-3p) was measured by quantitative real-time PCR using TaqMan® probes.” I think it would be more corrected to express the sentence in a more general term, and simply say that it has been measured the concentration of 3 circulating microRNAs rather that their “abundance”. The fact that the Author have found an abundant expression of these biomarkers should be then explained in the results.

(*Again, if the term “abundance” is not an adjective but a “technical word” used by experts of the miRNA analysis, Authors can ignore this Reviewer’s comment).

-line 32: “miR-30b-5p…” Not sure it is allowed to start a sentence with an abbreviation. Please, check rules of the Journal and eventually correct here and later in the text. For example, the same occurs at lines 38 and 52 (moreover, at line 32 and 52, the letter “m” is written with lowercase).

-Lines 38-39: “miR-30b-5p changed in the plasma of dogs at the asymptomatic stage of disease, particularly at a young age.” This last sentence is very concise. Probably, to highlight the potential clinical value of the study’s findings, I would expand it. If this is limited by the abstract’s word count, I probably would save words by reducing other parts of Abstract.

Introduction

-Lines 42-43: “The disease causes about 10% of all the deaths in this species [1]”. I have some perplexities about the reference that has been selected for this sentence. The study from reference 1 is not a research purposefully investigating the natural history, prognosis and rate of death of dogs with MMVD, but rather a dissertation about the link between the size of patients (and the pertinent genetic factors) and the predisposition to this valvular disease. Moreover, I am not able to find a sentence were it is specifically declared that the disease causes a rate of death of 10% in the affected dogs. A more appropriate reference should be selected (or, alternatively, some sentence’s modifications should be performed).

-Lines 43-44: “Although MMVD seems to be a genetic disorder, a mutation has not yet been identified”. The introduction of a study should be as complete as possible for readers. Although the aforesaid sentence is correct, it is very concise, especially concerning the pertinent reference. Since there has been, to date, several studies investigating the possible presence of genetic abnormalities in dogs affected by MMVD, I would introduce additional references. Otherwise the risk, for readers, could be that of thinking that only one study [ref. 2] has been performed on this topic.

-Lines 48-49: “Due to the lack of early signs, symptoms…” Which is the difference between signs and symptoms? Does “signs” stay for clinical signs? In that a case, signs and symptoms would represent a repetition. Please, clarify. Moreover, in this part of introduction, a brief citation of echocardiography is important. The point is that Authors should clarify to readers why the use a new biomarker could be useful. Indeed, a comment at regard could be that use of echocardiography is enough to identify canine MMVD since its very early stages (as the echocardiographic diagnosis is very easy). Probably, it could be useful to say that echo is the gold standard among non-invasive tests, but it requires expensive instrumentation and specific expertise; therefore, not all veterinarians could rely on such a test. Therefore, the availability of a reliable biomarker could be useful for some veterinarians in some specific contexts. Obviously this is just an example, authors are absolutely free to justify the need for such biomarker as they wish (the important thing is to contextualize it in the real veterinary word/small animal practice, where transthoracic echocardiography remains the most widely diffuse/used diagnostic tool).

-Lines 59-50: here the references from human and veterinary literature are melt together. I suggest to separate the two types of references to help readers to understand easily what refers to humans and what refers to dogs.

-Lines 63-65: “Most of the dogs enrolled in these studies were classified following American College of Veterinary Internal Medicine (ACVIM) guidelines as stage C and D, while only one study performed analysis also on dogs older than 8 years in ACVIM stages B1 and B2”. I have some perplexities about this sentence. Indeed, it does not appear completely correct to say that only 1 previous study has enrolled and studied dogs in the preclinical stage of MMVD as, to the Reviewer’s knowledge, this has been done in 4 studies (1 = BMC Vet Res. 2014 Sep 26;10:205. doi: 10.1186/s12917-014-0205-8. // 2 = Int J Mol Sci. 2015 Jun 19;16(6):14098-108. // 3 = J Extracell Vesicles. 2017 Jul 12;6(1):1350088. doi: 10.1080/20013078.2017.1350088. // 4 = Front Vet Sci . 2021 Oct 11;8:729929. doi: 10.3389/fvets.2021.729929. eCollection 2021.). Then, if for Authors a key point of this part of discussion concerns more the age of dogs rather their ACVIM class, it should be explained to readers why age is important. For example, does age influence miRNA expression in dogs? If this information in not available in dogs, are there similar data in other animal species? If the veterinary literature is completely free form researches at regard, it has been demonstrated in humans a role for age?

-Lines 67-68: “…by ascertaining how three miRNAs previously associated with MMVD…” Here it is important to put pertinent references, so that readers could easily know to which studies and which miRNAs Authors are referring to.

-Lines 68-69: “…are modulated in the plasma of CKCSs divided according to their age at the time of diagnosis (younger than 3 years, between 3 and 7 years, and older than 7 years).” I have some perplexities about this sentence. First, usually, the specific characteristics of each category is explained in the M&M section, not in the introduction. Nevertheless, if you want to maintain the specification of the 3 age categories here, it should at least explain why you specifically selected them. Are these age categories based on some biological criteria? Or is this just a personal/empirical/arbitrary decision?

-Line 70-79: The actual location of the sentences sounds somewhat strange to me. I think that this section, with some mild changes, could/should be moved before, for example at the end of the line 65. Indeed, there, Authors could say that, since previous study did not specifically focus exclusively on the early stage of the disease (indeed, the population of previous studies were very heterogeneous), they wanted to performed a more specific research; and then they could/should explain why they felt important to study exclusively dogs at stage ACVIM B1.

-Lastly, at the end of the abstract, it should be written which was the Authors/study’s hypothesis.

M&M

-Line 82: “The study included 44 owned CKCSs visited…” I suggest to delete the number of dogs as, actually, it is a result and in this section results should be not anticipated.

-Line 89-92: “The cardiovascular system was evaluated by checking the presence/absence of murmurs by two different well-trained operators, respectively a third year PhD student in cardiology and a professor with more than twenty years 91 of practice in clinical veterinary cardiology (MB and PGB).” I do not believe that the two types of operators involved in this study could be defined in an identical way/by an identical definition, as the experience of the second one (a professor with more than 25 years of experience) is significantly greater than that of the first one (a PhD student). Probably, it could be more correct to say “…by two operators with different levels of expertise….”.

-Lines 95-96: “Blood pressure was indirectly measured with a Doppler method according to the ACVIM consensus statement [31,32].” I suggest to maintain only the more recent reference; two references for this sentences seem excessive.

-Line 104: “The exam was performed by a certified cardiologist (MB)”. If I am not wrong, this author has been previously defined to be a PhD student. What “certified cardiologist” stays for?

-Line 121-123: “Mitral regurgitation was considered as trivial when the regurgitant jet was not detectable in all systolic events, while it was considered as trace when it was always visible [35].” The Reviewer really appreciate such a specification, the distinction between “trivial” and “trace” and the use of pertinent human literature. This attention is not always present in veterinary literature, and is very great to see this in the present study. Nevertheless, for a better reading, I would suggest to start the sentence with: “More in detail, mitral regurgitation was considered…”

-Additional comments: it is not completely clear which types of blood works have been conducted to be sure that dogs were really free from comorbidities (e.g., CBC and serum biochemistry?). These tests have been performed only in ACVIM A, only in B1, or in both? Moreover, did the presence of laboratory abnormalities represent an exclusion criterion only for ACIM A, or for both A and B1? Lastly, although according to current scientific evidence, no medication should be prescribed to MMVD at stage A and B1, unfortunately, it is not rare to find B1 dogs receiving a cardiovascular drug (like, for example, an ACE-inhibitor). Moreover, there are dogs receiving other medications (not cardiovascular), or dogs that receive specific diets that may affect cardiac function. Did the use of any drug represent another exclusion criterion? And what about particular diets (e.g., the grain-free ones)?

Results

-Table 1: Concerning the abbreviation, I suggest to convert, both in Table and in its legend, “LVIDas” and “LVIDad” into the more usual “LVIDs” and “LVIDd” by deleting the letter “a”.

-Line 202: “In detail, the abundance of miR-30b-5p increased 2.4 folds…” Again, can you confirm that the term “abundance” it more appropriate than the term “concentration”? I’m asking this also because, at line 206, Authors use the term “amount”. Probably a more generic term in the manuscript, for example like “concentration”, may reduce the risk of confusion for readers.

Discussion

-Lines 246-248: “… in asymptomatic MMVD-affected CKCSs (ACVIM stage B1 without cardiac remodeling, or with remodeling changes not severe enough to meet the current guidelines criteria for ACVIM class B2)…” As the definition of stage ACVIM B1 has been already provided to readers, there is no need to repeat it.

-Lines 289-290: “…without heart murmurs, without clinical signs,…” Please rephrase; a heart murmur is, actually, a clinical sign.

-Additional comments: the current version of the discussion lacks of some points of discussion. Indeed, as the Author

hypothesize that this study may lay the basis for a better breeding program for CKCS breeders, to the Reviewer’s opinion, they should discuss some few additional points:

1) a dissertation about the role of echocardiography, which is largely diffuse and used in small animal practice, should be written, justifying which could be the potential additional advantage offered by the introduction of a new biomarker over echo.

2) If a justification for this could be the limited availability of echo in some centres, then it should be also discussed how complex/easy is the analysis of miRNA and if labs form small veterinary hospitals may efforts such analysis (both technically and economically)

3) As age is an important parameter for Authors, a dissertation about the possible role of age categories on miRNA analysis should be written, combining data from either human and veterinary literature.

Lastly, I would suggest a recheck of the abbreviations used in the texts according to the Journal rules.

6. PLOS authors have the option to publish the peer review history of their article (what does this mean?). If published, this will include your full peer review and any attached files.

Reviewer #1: No

Reviewer #2: No

---

## [Author Response · Author response to Decision Letter 0]

1 Jun 2022

PONE-D-22-07425

Circulating miR-30b-5p is up regulated in Cavalier King Charles Spaniels affected by early myxomatous mitral valve disease

The authors thank the Editor and the Reviewers for their thoroughly review of our study. We are very pleased about their appreciation of the work and their positive comments. Thank you very much.

We have carefully considered all Reviewers’ comments and have tried to address them whenever we felt this was appropriate. We feel that the quality of our manuscript has improved following the Reviewers’ comments and suggestions.

Best regards

Reviewers' comments:

Reviewer #1: The present study reports interesting clinical research data on a emerging topic in veterinary cardiology (i.e. circulating microRNAs in spontaneus cardiovascular diseases). The study is well conducted and the manuscript is well written. Please consider the comments below.

Thank you very much for your appreciation of the work, we are really pleased. Thank you for your precise and interesting comments. 

Abstract: Please revised the abstract based on the comments below.

M&M

Line 104 – “MB” is here identified as “certified cardiologist” and as “third year PhD student” at lines 90-91. Please uniform, or clarify the training of the operator (“diplomate, resident, PhD student”?).

Thank you very much for this comment. MB was a third year PhD student at the time of the inclusion of the subjects. She is now a PhD (she discussed her thesis in March 31). The term “certified” was related to the Fondazione Salute Animale training for the echocardiographic certification in Italy. Anyway, to not create confusion and to not use incorrect terms, we decided to define MB only as a third year PhD student. 

Lines 114-115 – Please correct the term “left ventricular normalized dimensions in diastole (LVIDdN)“ in to “left ventricular end diastolic diameter normalized for body weight (LVIDdN)” as used in the 2019 ACVIM guidelines. And specify if M-mode or 2D was used for this measurement, and which view was used.

Thank you for this comment. We have corrected this term as indicated by the ACVIM guidelines. 

Results

Table 1 – Data regarding ESVI and EDVI are reported, but these variables are not described in the M&M. Were ESVI and EDVI obtained using the M-mode (Teichholz formula) and the 2D (e.g. Simpson method)? Specify. In the first case (M-mode derived volumes), I would consider not to include in the study because normalized linear measurements (LVIDDN and LVISDN) are sufficiently informative and most commonly used nowadays in veterinary cardiology.

Thank you for this comment. We calculated ESVI and EDVI starting from M-mode measurements so, as you suggested, we have deleted these data from the Table 1. 

Table 1 – Please standardize the abbreviations “LVIDas” and “LVIDad” to most used terms: “(LVIDDN) normalized left ventricular internal diameter in diastole” and “(LVIDSN) normalized left ventricular internal diameter in systole”.

Thank you for this suggestion. We have changed the terms. 

Table 1 – “EF%”, please specify if it is M-mode or 2D derived in the M&M.

Thank you, we have added this information in the caption.

Table 2 – Please indicate the unit of measurement of the cut-off in the table.

Thank you very much for this comment. We did not specify the unit of measurement because miRNAs conventionally are not expressed in measurement units. 

I would recommend to also analyze if B1 dogs without a cardiac murmur have different circulating levels of miR-30b-5p in comparison to B1 dogs with a murmur (independently for age); describing a possible cutoff to predict the presence of a murmur.

Thank you very much for this interesting comment. In this study we have not evaluated this correlation. This will be the focus of another study that we are ending, and that included B1 subjects with a follow up. Through an inverse probability weighted analysis (IPW) (with an outcome model), we have already analyzed the correlation between the presence/intensity of the heart murmur and the miR-30b-5p concentration. The marginal effect of miR-30b-5p on the presence/intensity of the heart murmur is always positive but not significant. This means that the increase of miR-30b-5p is correlated with an increase of the probability of the category with the lower value (in our case, absence of the heart murmur, or soft heart murmurs). In general, miR-30b-5p seems to inhibit the increase of this response variable. 

However, we have analyzed the required correlation also with the data of the present study. There were no differences in the expression of miR-30b-5p for the different murmur intensity categories (P=0.28). Furthermore, with the Kruskal-Wallis analysis, the miR-30-5p expression does not vary among the heart murmur categories (P=0.32). We think that the implementation of the number of the included subjects will probably help us confirming the trend observed with the IPW analysis. 

Similarly, it could be interesting to describe a possible cutoff of miR-30b-5p in discriminating dogs with a regurgitant jet size of 0-1-2 grade versus 3-4 grade. Please consider this implementation of the study.

Thank you for this comment. The response is the same as the previous one. We included the regurgitant jet size as an ordinal variable along with the presence/intensity of the heart murmur, and we obtained the same results.

Lines 304-306 – The authors state: “The identification of dogs with early asymptomatic MMVD through the evaluation of miR-30b-5p could help the clinicians and the breeders to better focalized screening programs in this breed and to better select the breeders”. This sentence seems to suggest that, based on the present study, the use of miR-30b-5p should be consider for MMVD screening and breeding programs in CKCSs. Personally, I believe that the clinical application of the study results is relatively questionable at this time, because of the small sample size and the not excellent sensitivity of the test (i.e. around 80%). Please remember that screening test are expected to have the highest sensitivity as possible. Echocardiography remains the gold standard for screening and breeding programs (Pedersen HD, et al. Echocardiographic mitral valve prolapse in cavalier King Charles spaniels: epidemiology and prognostic significance for regurgitation. Vet Rec 1999. Birkegård AC, et al. Breeding Restrictions Decrease the Prevalence of Myxomatous Mitral Valve Disease in Cavalier King Charles Spaniels over an 8- to 10-Year Period. J Vet Intern Med. 2016). The results of the present study are interesting as new clinical research data on the disease. The possible clinical use of the miR-30b-5p for screening and breeding programs in CKCS needs stronger data.

Thank you very much for this interesting comment. We completely agree with your opinion. Probably this statement can be misunderstood, so, based on your comment, we modified it as follow: “The identification of dogs with early asymptomatic MMVD through the evaluation of miR-30b-5p is interesting as new clinical research data in this breed, and in the affected canine population. Obviously, the possible clinical use of the miR-30b-5p for screening and breeding programs in the CKCS needs stronger data, that could help the clinicians and the breeders to better focalize screening programs and to carefully select the breeders. Patients with these characteristics should then be subjected to a closer follow-up. For these reasons, miRNAs may be candidates as novel biomarkers and may provide the basis for further investigations, in order to assess the follow-up and characterize the evolution of the disease in the CKCS, without evading an echocardiographic evaluation, that is currently the gold standard for MMVD diagnosis”.

Discussion

Line 243 – Please verify the term “abundance”. “circulating levels” or “levels” maybe better?

Thank you very much for this comment. The coauthors CL and FC, experts in the field of miRNA analysis, confirmed the correct use of the term “abundance” in this statement, for this reason we decided not to change it. Thank you very much. 

Line 284 – Please verify the term “preventive”. “Screening” or “breeding” programs mays be better?

Thank you very much, we agree with this comment, and we deleted “preventive” from the text. 

Conclusions

The authors state “The present results lay the basis for a breeding program that will help the CKCS’ breeders in their targeted selection to obtain healthier subjects with a reasonable life expectancy”. Again, this conclusion is too strong in my personal opinion and can be a misleading message for the reader. Echocardiography is the non-invasive gold standard method for MMVD diagnosis, and the diagnostic accuracy of miR-30b-5p levels are good, but not enough to consider it a screening test. Considering the common breeding age of 1-4 years, a sensitivity of around 80% and a specificity of around 70% (described in the results) are relatively low for thinking using miR-30b-5p levels in the screening of subjects possibly used for reproduction. There is a relatively high risk of including affected dogs in the program. Personally, I suggest to reconsider the conclusion message.

Similarly, the authors state “…will favour a preventive screening and a mitigating therapeutical approach”. Again, I feel this conclusion as too strong based on the results of the study. Especially I would avoid any reference to “therapeutical” considerations.

Again, thank you very much for this comment. We probably wrote a misleading statement and for this reason we modified the section as follow: “The present results lay the basis for future studies aimed at reaching stronger data, that will help the CKCS’ breeders in their targeted selection programs, based on the echocardiographic evaluation, and to obtain healthier subjects with a reasonable life expectancy. At the same time, highlighting the risk of the development of the disease at an earlier stage will favour a focused screening of the subjects. To that end, the identification of early biomarkers for premature MMVD would be a helpful addition.”

Reviewer #2: The manuscript describes the use of a novel type of biomarker, namely microRNA, in dogs affected by myxomatous mitral valve disease. The study’s idea as well as the topic of research are innovative and brilliant. Similarly, a great work has been done by the Authors to combine clinical and cardiological expertise and techniques with laboratory ones. The findings described herein are interesting and the data reported may lay the fundaments for further studies on this topic. Given the above, it was a great opportunity for me to Review this manuscript, and I congratulate with Author for their great job. Below some comments, questions and suggestions aimed at expanding further the results of the study and provide additional information to readers.

We thank you a lot for your appreciation of the work, we are really pleased for the review and for your positive comments. Thank you for your many suggestions.

Abstract

-lines 22-23: “The aim of the study was to measure the abundance of 3 circulating microRNAs…” Probably, it could be deleted the word “abundance”, as it indirectly implies a result. At the beginning of a study on new field of veterinary medicine, it is almost impossible to be sure that a specific biomarker would be for sure “abundant” or “scant” in a specific population. Therefore, I think it would be more appropriate for the introduction of the abstract this type of sentence: “The aim of the study was to measure 3 circulating microRNAs…”; then, in the result section, it could be stated that “abundance” of the biomarker has been found in a part of the study population.

(*What has been written here, however, should not be considered by the Author if the term “abundance” is the one specifically proposed by the experts in the field of miRNA analysis, and if it is used by them not like an adjective but like a technical word).

Thank you very much for your comment, we completely agree. For this reason, we have deleted the term abundance from the abstract.

-line 25: “33 dogs affected…” Not sure it is allowed to start a sentence with a number. Please, check rules of the Journal and eventually correct here and later in the text.

Thank you, the journal guidelines did not report this rule, but we agree with the Reviewer, and we changed it. 

-line 29: “This is a prospective cross-sectional study”. The sentence that declares the type of study design typically is put before the description of the details of the study population. Accordingly, I would put it before the sentence starting with “33 dogs affected…”.

Thank you for the comment. We moved the sentence as suggested. 

-lines 29-31: “The abundance of three circulating microRNAs (miR-1-3p, miR30b-5p, and miR-128-3p) was measured by quantitative real-time PCR using TaqMan® probes.” I think it would be more corrected to express the sentence in a more general term, and simply say that it has been measured the concentration of 3 circulating microRNAs rather that their “abundance”. The fact that the Author have found an abundant expression of these biomarkers should be then explained in the results.

(*Again, if the term “abundance” is not an adjective but a “technical word” used by experts of the miRNA analysis, Authors can ignore this Reviewer’s comment).

Thank you, we agree with this comment and the previous one.

-line 32: “miR-30b-5p…” Not sure it is allowed to start a sentence with an abbreviation. Please, check rules of the Journal and eventually correct here and later in the text. For example, the same occurs at lines 38 and 52 (moreover, at line 32 and 52, the letter “m” is written with lowercase).

Thank you very much for this comment. We have checked the rules of the Journal and it is not specifically declared. However, we agree with the Reviewer, and we have corrected the first letter in capital font. We have also checked the other abbreviations in the text, and we have corrected them if they started a sentence, except for “RNA” abbreviation that is widely accepted. 

-Lines 38-39: “miR-30b-5p changed in the plasma of dogs at the asymptomatic stage of disease, particularly at a young age.” This last sentence is very concise. Probably, to highlight the potential clinical value of the study’s findings, I would expand it. If this is limited by the abstract’s word count, I probably would save words by reducing other parts of Abstract.

Thank you very much for this comment. We have changed the final sentence of the abstract to be less concise.

Introduction

-Lines 42-43: “The disease causes about 10% of all the deaths in this species [1]”. I have some perplexities about the reference that has been selected for this sentence. The study from reference 1 is not a research purposefully investigating the natural history, prognosis and rate of death of dogs with MMVD, but rather a dissertation about the link between the size of patients (and the pertinent genetic factors) and the predisposition to this valvular disease. Moreover, I am not able to find a sentence were it is specifically declared that the disease causes a rate of death of 10% in the affected dogs. A more appropriate reference should be selected (or, alternatively, some sentence’s modifications should be performed).

Thank you very much for this comment, we agree whit you, probably it was a mistake. We have deleted the sentence and we have inserted this paper as reference 1: 

Borgarelli M, Buchanan JW. Historical review, epidemiology and natural history of degenerative mitral valve disease. J Vet Cardiol. 2012;14: 93-101.

-Lines 43-44: “Although MMVD seems to be a genetic disorder, a mutation has not yet been identified”. The introduction of a study should be as complete as possible for readers. Although the aforesaid sentence is correct, it is very concise, especially concerning the pertinent reference. Since there has been, to date, several studies investigating the possible presence of genetic abnormalities in dogs affected by MMVD, I would introduce additional references. Otherwise the risk, for readers, could be that of thinking that only one study [ref. 2] has been performed on this topic.

Thank you. We agree, only one reference is not enough, so we have modified the sentence and added other references:

- Lewis T, Swift S, Woolliams JA, Blott S. Heritability of premature mitral valve disease in Cavalier King Charles spaniels. Vet J. 2011;188(1): 73-76. 

- Meurs KM, Friedenberg SG, Williams B, Keene BW, Atkins CE, Adin D, Aona B, DeFrancesco T, Tou S, Mackay T. Evaluation of genes associated with human myxomatous mitral valve disease in dogs with familial myxomatous mitral valve degeneration. Vet J. 2018 Feb;232:16-19. 

-Lines 48-49: “Due to the lack of early signs, symptoms…” Which is the difference between signs and symptoms? Does “signs” stay for clinical signs? In that a case, signs and symptoms would represent a repetition. Please, clarify. 

Thank you for the comment. Basing on human medicine, we speak of symptom as something subjective, perceived by the patient using the senses, and of sign as an abnormality objectively interpreted by the doctor as an index of disease. So, we decided to maintain the term “early clinical signs”.

Moreover, in this part of introduction, a brief citation of echocardiography is important. The point is that Authors should clarify to readers why the use a new biomarker could be useful. Indeed, a comment at regard could be that use of echocardiography is enough to identify canine MMVD since its very early stages (as the echocardiographic diagnosis is very easy). Probably, it could be useful to say that echo is the gold standard among non-invasive tests, but it requires expensive instrumentation and specific expertise; therefore, not all veterinarians could rely on such a test. Therefore, the availability of a reliable biomarker could be useful for some veterinarians in some specific contexts. Obviously this is just an example, authors are absolutely free to justify the need for such biomarker as they wish (the important thing is to contextualize it in the real veterinary word/small animal practice, where transthoracic echocardiography remains the most widely diffuse/used diagnostic tool).

Thank you, we have added this information. Thank you very much for your suggestion. The paragraph has been modified as follow:

“Transthoracic echocardiography is currently considered the gold standard for the diagnosis of MMVD. However, this test needs specialized equipment and well-trained operators to reduce interobserver variability, as valves affected by mild changes may be interpreted as normal. For this reason, identifying reliable specific biomarkers is desirable, especially for screening and breeding programs.”

-Lines 59-50: here the references from human and veterinary literature are melt together. I suggest to separate the two types of references to help readers to understand easily what refers to humans and what refers to dogs.

Thank you, we have divided the references as you suggested. 

-Lines 63-65: “Most of the dogs enrolled in these studies were classified following American College of Veterinary Internal Medicine (ACVIM) guidelines as stage C and D, while only one study performed analysis also on dogs older than 8 years in ACVIM stages B1 and B2”. I have some perplexities about this sentence. Indeed, it does not appear completely correct to say that only 1 previous study has enrolled and studied dogs in the preclinical stage of MMVD as, to the Reviewer’s knowledge, this has been done in 4 studies (1 = BMC Vet Res. 2014 Sep 26;10:205. doi: 10.1186/s12917-014-0205-8. // 2 = Int J Mol Sci. 2015 Jun 19;16(6):14098-108. // 3 = J Extracell Vesicles. 2017 Jul 12;6(1):1350088. doi: 10.1080/20013078.2017.1350088. // 4 = Front Vet Sci . 2021 Oct 11;8:729929. doi: 10.3389/fvets.2021.729929. eCollection 2021.). 

Thank you for this comment, we agree with the reviewer, we have modified the sentence and we have added other references related to the evaluation of miRNAs expression in dogs in ACVIM stage B1 and B2 (or in general class B-asymptomatic). 

Then, if for Authors a key point of this part of discussion concerns more the age of dogs rather their ACVIM class, it should be explained to readers why age is important. For example, does age influence miRNA expression in dogs? If this information in not available in dogs, are there similar data in other animal species? If the veterinary literature is completely free form researches at regard, it has been demonstrated in humans a role for age?

Thank you for this comment. The key point of the sentence did not concern the age of the dogs. However, the reviewer had highlighted an interesting point. Human medicine literature reports some studies about this interesting argument. Age-related changes in miRNA expression may be one mechanism regulating gene expression at different developmental stages. Only a few studies have examined age-associated differences in miRNA expression in children. Over 100 liver miRNAs (out of 533) were differentially expressed among fetal, pediatric, and adult subjects, with the largest differences observed between fetal and pediatric liver samples [Burgess et al. Age-Related Changes in MicroRNA Expression and Pharmacogenes in Human Liver. Clin. Pharmacol. Ther. 2015;98:205–215. doi: 10.1002/cpt.145.]. Among adults, most miRNAs expressed in peripheral blood have exhibited lower expression levels in older people [Noren Hooten et al. microRNA Expression Patterns Reveal Differential Expression of Target Genes with Age. PLoS ONE. 2010;5:e10724. doi: 10.1371/journal.pone.0010724.]. While studies on the relationship between age and sex and miRNA expression are still limited, the existing data suggest that consideration of important host factors is critical for epidemiological studies of miRNA expression. Furthermore, MiRNAs play a fundamental role in regulating the processes underlying these detrimental changes in the human cardiac system, indicating that MiRNAs are crucially involved in aging. Among others, MiR-34, MiR-146a, and members of the MiR-17-92 cluster, are deregulated during senescence, and drive cardiac aging processes. It is therefore suggested that MiRNAs form possible therapeutic targets to stabilize the aged failing myocardium [Verjans et al. MiRNA Deregulation in Cardiac Aging and Associated Disorders. Int Rev Cell Mol Biol. 2017;334:207-263. doi: 10.1016/bs.ircmb.2017.03.004]. Veterinary medicine literature lacks this type of information, particularly regarding the mitral valve disease. For these reasons, we think that veterinary medicine studies will be needed to understand the relationship between age and miRNAs expression in dogs affected by mitral valve disease and in other diseases (like oncologic or metabolic ones). 

We decided to not add this information in the text and to delete the reference of the age (8 years) to not create misunderstandings. Thank you. 

-Lines 67-68: “…by ascertaining how three miRNAs previously associated with MMVD…” Here it is important to put pertinent references, so that readers could easily know to which studies and which miRNAs Authors are referring to.

Thank you, this is a forgetfulness. We have added the correct references.

-Lines 68-69: “…are modulated in the plasma of CKCSs divided according to their age at the time of diagnosis (younger than 3 years, between 3 and 7 years, and older than 7 years).” I have some perplexities about this sentence. First, usually, the specific characteristics of each category is explained in the M&M section, not in the introduction. Nevertheless, if you want to maintain the specification of the 3 age categories here, it should at least explain why you specifically selected them. Are these age categories based on some biological criteria? Or is this just a personal/empirical/arbitrary decision?

Thank you for the comment. We agree with the Reviewer, so we have deleted the information about the three age classes from the introduction. Furthermore, we have declared that the classification was based on an arbitrarily decision made after considering literature data. We have also added the references. 

-Line 70-79: The actual location of the sentences sounds somewhat strange to me. I think that this section, with some mild changes, could/should be moved before, for example at the end of the line 65. Indeed, there, Authors could say that, since previous study did not specifically focus exclusively on the early stage of the disease (indeed, the population of previous studies were very heterogeneous), they wanted to performed a more specific research; and then they could/should explain why they felt important to study exclusively dogs at stage ACVIM B1.

Thank you for this comment. We have moved the sentences at ex-line 65 and we have also underlined that the previous studies did not specifically focus on the early stages of the disease. We highlighted the variability of the phenotypic characteristics of subjects in ACVIM class B1 and stressed that the disease sometimes goes undetected. This often happens in subjects that have no clinical signs and that do not present heart murmurs at the visit, thus are not subjected to an echocardiographic evaluation.

-Lastly, at the end of the abstract, it should be written which was the Authors/study’s hypothesis.

Thank you for this suggestion. We have added our hypothesis in the abstract. We decided not to add this part at the end but in the middle part of the abstract, to help the reader to better understand our goals. 

M&M

-Line 82: “The study included 44 owned CKCSs visited…” I suggest to delete the number of dogs as, actually, it is a result and in this section results should be not anticipated.

Thank you for this comment. We have deleted this data from the M&M section.

-Line 89-92: “The cardiovascular system was evaluated by checking the presence/absence of murmurs by two different well-trained operators, respectively a third year PhD student in cardiology and a professor with more than twenty years 91 of practice in clinical veterinary cardiology (MB and PGB).” I do not believe that the two types of operators involved in this study could be defined in an identical way/by an identical definition, as the experience of the second one (a professor with more than 25 years of experience) is significantly greater than that of the first one (a PhD student). Probably, it could be more correct to say “…by two operators with different levels of expertise….”.

Thank you. We agree with the Reviewer, and we changed the sentence as suggested.

-Lines 95-96: “Blood pressure was indirectly measured with a Doppler method according to the ACVIM consensus statement [31,32].” I suggest to maintain only the more recent reference; two references for this sentences seem excessive.

Thank you for this comment, we agree. We have deleted the oldest reference. Thank you. 

-Line 104: “The exam was performed by a certified cardiologist (MB)”. If I am not wrong, this author has been previously defined to be a PhD student. What “certified cardiologist” stays for?

Thank you very much for this comment. The same perplexity was exposed by Reviewer 1. MB was a third year PhD student at the time of the inclusion of the subjects. She is now a PhD (she discussed her thesis in March 31). The term “certified” was related to the Fondazione Salute Animale training for the echocardiographic certification in Italy. Anyway, to not create confusion and to not use incorrect terms, we decided to define MB only as a third year PhD student. 

-Line 121-123: “Mitral regurgitation was considered as trivial when the regurgitant jet was not detectable in all systolic events, while it was considered as trace when it was always visible [35].” The Reviewer really appreciate such a specification, the distinction between “trivial” and “trace” and the use of pertinent human literature. This attention is not always present in veterinary literature, and is very great to see this in the present study. Nevertheless, for a better reading, I would suggest to start the sentence with: “More in detail, mitral regurgitation was considered…”

Thank you very much for your appreciation regarding the classification of the regurgitant jet size, we think that this is an interesting aspect to consider in the description of B1 subjects. We have modified the sentence according to your suggestion. Thank you.

-Additional comments: it is not completely clear which types of blood works have been conducted to be sure that dogs were really free from comorbidities (e.g., CBC and serum biochemistry?). These tests have been performed only in ACVIM A, only in B1, or in both? Moreover, did the presence of laboratory abnormalities represent an exclusion criterion only for ACIM A, or for both A and B1? 

Thank you for these comments and requests. We have added this information in the text. We have performed routine bloodwork (i.e., complete blood cell count and serum biochemistry) to rule out possible comorbidities in both A and B1 subjects. All the included subjects had normal hematobiochemical exams. For future studies, surely, any type of abnormality will represent an exclusion. This is very important because we can suppose, basing on published data, that miRNAs expression levels may vary if other comorbidities are present, and, at this level of research, we needed to be sure that nothing of what we can rule out was influencing our results. 

Lastly, although according to current scientific evidence, no medication should be prescribed to MMVD at stage A and B1, unfortunately, it is not rare to find B1 dogs receiving a cardiovascular drug (like, for example, an ACE-inhibitor). Moreover, there are dogs receiving other medications (not cardiovascular), or dogs that receive specific diets that may affect cardiac function. Did the use of any drug represent another exclusion criterion?

Thank you for this comment. Fortunately, this is a prospective study and we have included only B1 dogs that had never received any cardioactive therapy or other drugs in the previous 6 months. We have added this information in the materials and methods section. Thank you for having induced us to add this important clue. 

And what about particular diets (e.g., the grain-free ones)?

Thank you a lot. That’s a great question and a very interesting topic. We have considered it ourselves, because we believe that diet, as already demonstrated in some works published by Nestlè Purina and by veterinary nutritionists, can influence the progression of the disease, providing nutrient mixtures that can slow down the cardiac damage. Unfortunately, when we included the subjects in our study we did not take this into account. But we have now specified it in the text.

In addition, grain-free diets are under great discussion, especially in some breeds (such as the Golden retriever or other giant breeds) predisposed to the development of dilated cardiomyopathy and taurine deficiency. There are no literature reports that describe the influence of grain-free diet on the development or progression of mitral valve disease in dogs, but some papers report the influence of these diets on cardiac biomarkers (Adin D, Freeman L, Stepien R, Rush JE, Tjostheim S, Kellihan H, Aherne M, Vereb M, Goldberg R. Effect of type of diet on blood and plasma taurine concentrations, cardiac biomarkers, and echocardiograms in 4 dog breeds. J Vet Intern Med. 2021 Mar;35(2):771-779. doi: 10.1111/jvim.16075. Epub 2021 Feb 27. PMID: 33638176; PMCID: PMC7995416) and we are very interested in the evaluation of miRNAs changes in relation to diet. We believe that this will be an interest topic to deepen, and it is certainly a starting point for our future research. 

If you are interested, these are the articles we referred to before: 

- Li Q, Freeman LM, Rush JE, Huggins GS, Kennedy AD, Labuda JA, Laflamme DP, Hannah SS. Veterinary Medicine and Multi-Omics Research for Future Nutrition Targets: Metabolomics and Transcriptomics of the Common Degenerative Mitral Valve Disease in Dogs. OMICS. 2015 Aug;19(8):461-70. doi: 10.1089/omi.2015.0057. Epub 2015 Jul 8. PMID: 26154239.

- Li Q, Heaney A, Langenfeld-McCoy N, Boler BV, Laflamme DP. Dietary intervention reduces left atrial enlargement in dogs with early preclinical myxomatous mitral valve disease: a blinded randomized controlled study in 36 dogs. BMC Vet Res. 2019 Nov 27;15(1):425. doi: 10.1186/s12917-019-2169-1. PMID: 31775756; PMCID: PMC6882217.

- Li Q, Laflamme DP, Bauer JE. Serum untargeted metabolomic changes in response to diet intervention in dogs with preclinical myxomatous mitral valve disease. PLoS One. 2020 Jun 18;15(6):e0234404. doi: 10.1371/journal.pone.0234404. PMID: 32555688; PMCID: PMC7302913.

Results

-Table 1: Concerning the abbreviation, I suggest to convert, both in Table and in its legend, “LVIDas” and “LVIDad” into the more usual “LVIDs” and “LVIDd” by deleting the letter “a”.

Thank you for this comment, as suggested by Reviewer 1 we have modified the term as “LVIDDN and LVIDSN, normalized left ventricular internal diameter in diastole and systole”.

-Line 202: “In detail, the abundance of miR-30b-5p increased 2.4 folds…” Again, can you confirm that the term “abundance” it more appropriate than the term “concentration”? I’m asking this also because, at line 206, Authors use the term “amount”. Probably a more generic term in the manuscript, for example like “concentration”, may reduce the risk of confusion for readers.

Thank you very much for this comment. We did not use the term concentration because it requires a unit of measurement (e.g. mg/ml); since we have carried out a relative quantification, expressed in fold changes, the terms ‘amount and level’ are more correct. 

Discussion

-Lines 246-248: “… in asymptomatic MMVD-affected CKCSs (ACVIM stage B1 without cardiac remodeling, or with remodeling changes not severe enough to meet the current guidelines criteria for ACVIM class B2)…” As the definition of stage ACVIM B1 has been already provided to readers, there is no need to repeat it.

Thank you for this suggestion, we have deleted this information.

-Lines 289-290: “…without heart murmurs, without clinical signs,…” Please rephrase; a heart murmur is, actually, a clinical sign.

Thank you for this comment, we completely agree and so we have modified the sentence as: “without audible heart murmurs and other clinical signs”.

-Additional comments: the current version of the discussion lacks of some points of discussion. Indeed, as the Author

hypothesize that this study may lay the basis for a better breeding program for CKCS breeders, to the Reviewer’s opinion, they should discuss some few additional points:

1) a dissertation about the role of echocardiography, which is largely diffuse and used in small animal practice, should be written, justifying which could be the potential additional advantage offered by the introduction of a new biomarker over echo.

Thank you for this comment. The original assumption of the present research project is that the diagnostic approach to canine MMVD is still affected by some shortcomings, especially concerning the ability to detect the very early stage of the disease, and that the selection of dogs of highly predisposed breeds, such as CKCSs, deserves more attention to reduce the diffusion of this condition. Our assumption intrinsically paves the way towards exploring new diagnostic strategies that may represents alternative/complementary tools to the traditional ones (such as echocardiography). Considering the hereditary nature of MMVD, a possibility aimed at optimizing the genetic selection of breeders and creating genealogies composed predominantly by healthy individuals could have been to design a research protocol based on molecular genetics. Indeed, theoretically, such an advanced laboratory approach may be able to elucidate the hereditary mechanisms of canine MMVD. Moreover, it is important to underline that molecular genetic tests are extremely expensive and complex to carry out; therefore, only some highly specialized laboratories can undertake this type of tests and bear the relative costs. For these reasons, the use of such a diagnostic approach in the daily small animal practice and for the screening of dogs belonging to breeds predisposed to MMVD is hardly conceivable, if not excludable at all. Given the above, this project has been designed to contribute providing a different screening modality for CKCSs affected by MMVD, which may be not only highly reliable, but also usable by a wider audience. In other words, our aim was to further contribute exploring the pathophysiological mechanisms/pathways of gene expression/modulation in canine MMVD by using a technology that is more accessible than traditional molecular genetic tests. For these reasons, we aimed at exploring the potential application of miRNAs analysis, with the final goal of supporting breeders in their targeted selection programs to obtain healthier subjects; this, in turn, would ensure the protection of the genetic pool of the breed, which represents an important national and international goal.

For these reasons, we have added this dissertation in the text, as you suggested: 

“Our results may pave the way towards incorporating this new generation of biomarkers in the traditional diagnostic approach, currently based simply on physical and echocardiographic examinations, to achieve a prompter and more accurate identification of affected dogs. This, in turn, may contribute to treat appropriately the disease since its early stages and, consequently, decrease the mortality rate of MMVD in CKCSs. Especially this last result, if confirmed, would be revolutionary as it would radically change the prognostic perspectives of the most widespread canine heart disease and the most common cause of death in many small-sized canine breeds.”

2) If a justification for this could be the limited availability of echo in some centres, then it should be also discussed how complex/easy is the analysis of miRNA and if labs form small veterinary hospitals may efforts such analysis (both technically and economically)

Thank you for this comment. This type of screening cannot be proposed on a large scale of veterinary hospitals, but we believe that, if the results will be confirmed with the follow up of the included subjects (we are already working on this) and with a larger number of dogs, a kit could be patented, and maybe a rapid test for the evaluation of miRNAs level can be created and widely commercialized. Unfortunately, we are still far from this reality, but we are working to improve our results. For this reason, we have added in the text this speculation: 

“Findings from the present research may open a florid collaboration with biomedical companies in order to develop rapid in-clinic/at-home diagnostic devices aimed at evaluating accurately, rapidly, and easily the expression of selected miRNAs. This would further strengthen the technical collaboration between clinicians and biotechnologists (leading to a reciprocal scientific enrichment) and would have a potential positive economic impact for involved biomedical companies.”

3) As age is an important parameter for Authors, a dissertation about the possible role of age categories on miRNA analysis should be written, combining data from either human and veterinary literature.

A human study published in 2019 [Huen et al. Age-Related Differences in miRNA Expression in Mexican-American Newborns and Children. Int J Environ Res Public Health. 2019;16(4):524. doi:10.3390/ijerph16040524] affirmed that, unlike adults, where miRNA expression levels in peripheral blood may decrease with age, expression levels of most miRNAs increased from birth to mid-childhood. Pathway analysis revealed several ontology terms related to post-translational modifications, metabolism, and immune response that were enriched among predicted targets of age-associated miRNAs. Given that the majority of differentially expressed miRNAs were upregulated at age seven (versus newborns), the predicted mRNA targets and their associated pathways highlighted processes important in biological development as children get older.

This may be reflective of the role miRNAs may play in the highly coordinated mechanisms regulating genes involved in children’s development and maybe also in pets and cattle. Obviously we need to create ad hoc veterinary studies in which the age categories are correctly classified also according to the breed and/or species (for example, a small medium-sized dog at 1 year can be considered an adult dog, the same cannot be said for a giant dog or for cattle). Furthermore, it will be important to adjust for age, sex, and blood cell composition the miRNA expression in blood in future veterinary studies.

We have added a statement in the discussion. Thank you for this interesting suggestion. 

Lastly, I would suggest a recheck of the abbreviations used in the texts according to the Journal rules.

Thank you for this suggestion, we have rechecked all the abbreviations used in the text according to the Journal rules. Thank

---

## [Decision Letter · Decision Letter 1]

16 Jun 2022

PONE-D-22-07425R1Circulating miR-30b-5p is up regulated in Cavalier King Charles Spaniels affected by early myxomatous mitral valve diseasePLOS ONE

Dear Dr. Brambilla,

Thank you for submitting your manuscript to PLOS ONE. After careful consideration, we feel that it has merit but does not fully meet PLOS ONE’s publication criteria as it currently stands. Therefore, we invite you to submit a revised version of the manuscript that addresses the points raised during the review process.

We look forward to receiving your revised manuscript.

Kind regards,

Vincenzo Lionetti, M.D., PhD

Academic Editor

PLOS ONE

Journal Requirements:

Reviewers' comments:

Reviewer's Responses to Questions

**Comments to the Author**

1. If the authors have adequately addressed your comments raised in a previous round of review and you feel that this manuscript is now acceptable for publication, you may indicate that here to bypass the “Comments to the Author” section, enter your conflict of interest statement in the “Confidential to Editor” section, and submit your "Accept" recommendation.

Reviewer #1: All comments have been addressed

Reviewer #2: All comments have been addressed

2. Is the manuscript technically sound, and do the data support the conclusions?

Reviewer #1: Yes

Reviewer #2: Yes

3. Has the statistical analysis been performed appropriately and rigorously? 

Reviewer #1: Yes

Reviewer #2: Yes

4. Have the authors made all data underlying the findings in their manuscript fully available?

Reviewer #1: Yes

Reviewer #2: Yes

5. Is the manuscript presented in an intelligible fashion and written in standard English?

Reviewer #1: Yes

Reviewer #2: Yes

6. Review Comments to the Author

Reviewer #1: Thanks to the Authors for the work done in improving the manuscript, that can now be accepted for publication in this reviewer's opinion.

Reviewer #2: I really appreciate the changes made by the Authors and the additional information provided. Such modifications increase the quality of the manuscript and the clinical relevance of its results. I thank the Authors for their great work. However, I have some residual comments that have been listed below. The majority of them are related to the way the sentences are structured and interconnected, especially in the Abstract (which is still a little bit complex to be read and may create some confusion to readers).

Abstract

-lines 39: the word “abundance” I still present. As this term sounds a little be ambiguous, I would recommend to change it with another word, such as “amount” or a synonym.

-last 3 lines of the abstract: I would suggest to modify the end of the abstract (current version “Identifying dogs with early asymptomatic myxomatous mitral valve disease through the evaluation of miR-30b-5p as new clinical research data in this breed supports more focused screening programs”) in this way (or something similar): “Identifying dogs with early asymptomatic myxomatous mitral valve disease through the evaluation of miR-30b-5p represents an intriguing possibility that certainly merits further research. Studies enrolling a larger number of dogs with preclinical stages of MMVD are needed to expand further and validate conclusively the preliminary findings from this report”.

Introduction

-Lines 67: I would suggest the word “interpreted” with the term “misinterpreted” (so that the sentence would become: “However, this test needs specialized equipment and well-trained operators to reduce interobserver variability, as valves affected by mild changes may be misinterpreted as normal”).

-Lines 77-85: I would suggest to change the sentences included in these lines in this way (or something similar): “Aberrant expression of miRNAs is associated with several human [18-21] and veterinary [22-25] disorders, including cancer and heart diseases. With specific regard to canine MMVD, the dysregulation of circulating miRNAs has been previously investigated by different approaches, including real-time quantitative PCR, microarray, and next-generation sequencing [26-31]. However, it should be noticed that most of the dogs enrolled in these studies were classified as American College of Veterinary Internal Medicine (ACVIM) stages C and D, while only one previous study has investigated circulating miRNAs in adult dogs with preclinical MMVD (i.e., ACVIM stages B1 and B2).”

-Line 89-93: I would suggest to change the sentences included in these lines in this way (or something similar): “It should be also highlighted that data from previous researches on dysregulation of circulating miRNAs in dogs with MMVD are partially biased by the limited sample sizes and the heterogeneity of the therapeutic protocols as well as study populations, as they were not specifically focused exclusively on one specific stage of the disease [28,29,31,33,34]”.

-Lines 93-98: I would suggest to change the sentences included in these lines in this way (or something similar): “Given the above, this study was aimed at investigating the potential use of miRNA as biomarkers to identify dogs belonging to a specific ACVIM class, namely the stage B1. The decision to focus on this ACVIM class was driven by the fact that these dogs are most subjected to breed screening, and therefore are targeted as potential breeders. Moreover, the inclusion of a single ACVIM class offered the advantage of investigating miRNA in a very homogenous study population, which, in this case, is also free from the potential confounding effects of cardiovascular drugs (as ACVIM stage B1 do not need medical treatment for their underlying structural heart disease [reference of ACVIM guidelines 2019])”.

-Lines 98-104: I would suggest to deleted sentences included in these lines as the sound a little bit repetitive and not very useful in light of the topic and aim of the study (from “However, the disease goes undetected in subjects….” to “….will favour a focused screening of the subjects”).

-Lines 106-110: I would suggest to modified these lines in this way (or something similar): “To further standardize our study population we also selected a specific canine breed, namely the CKCS, being the most commonly affected by MMVD [pertinent references]. Lastly, we focused our analysis on three specific miRNAs previously demonstrated to be associated with canine MMVD [29,31,33] and arbitrarily divided our study population in three distinct age categories (see below for further details) to explore how these miRNAs are modulated in the plasma of CKCS of different ages affected by MMVD at stage ACVIM B1”.

Discussion

-Lines 343-345: I have some perplexity about this sentence: “We demonstrated that miR-30b-5p could discriminate among ACVIM stage A CKCSs and ACVIM stage B1 CKCSs younger than 3 years, without audible heart murmurs and other clinical signs.” Specifically, my perplexity is about the “other clinical signs”. Indeed, dogs with MMVD at stage B1 do no have clinical signs at all, apart from the typical left-sided systolic heart murmur. Therefore, I would suggest deleting “and other clinical signs”.

-Lines 370-371: I would suggest to modify the sentence in this way: “without evading the echocardiographic evaluation, which undoubtfully remains the gold standard for MMVD diagnosis”.

-Line 375: correct the current version: you can write “contribute to treat” or “contribute treating”, while contribute to treating sounds wrong.

-Line 376: replace the word “consequently” with the term “hopefully” or “theoretically”.

-Lines 377-378: Modify the sentence, so that the final result would be “If confirmed, this last result would be revolutionary as it would radically change the prognostic perspectives of…”

-Lines 385-386: I would further underline the fact that, currently, a widely available, rapid, reliable, not expensive kit for miRNAs analysis is lacking. Therefore, at the end of the sentence at line 386, I would write something similar to this: “Indeed, it should be note that, to date, a rapid, not expensive, widely available diagnostic kit to measure the miRNAs amount is not available in small animal practice; therefore, such an analysis is currently limited to few highly specialized laboratories”.

-Lines 406-410: I would suggest to change these sentences in this way (or something similar): “Furthermore, in human medicine it has been demonstrated a potential link between patient’s age and miRNAs profile. For example, it has been showed that miRNAs may play a role in the highly coordinated mechanisms regulating genes involved in children’s development [55]. Regrettably, in veterinary medicine, to date, such a link has not been investigated. For this reason, veterinary studies purposefully designed to take into account subjects’ age categories and investigate the link between age and miRNAs profile are needed”.

Conclusions

I believe that the conclusions of the study are overall out of topic. A study’s conclusion should primarily resume the key points relates to the main results of the study, so that readers can have some concise, clear take of message of the current research. In contrast, conclusions should not include speculations about the future and the possible, theorical application on the results of future studies which will be based on the present manuscript. Obviously, a sentence on the future prospective is desirable, but it should be a minimal part of the conclusion section, not its predominant/entire one. Therefore, I kindly invite Authors to rewrite their conclusions.

7. PLOS authors have the option to publish the peer review history of their article (what does this mean?). If published, this will include your full peer review and any attached files.

Reviewer #1: No

Reviewer #2: No

---

## [Author Response · Author response to Decision Letter 1]

20 Jun 2022

PONE-D-22-07425R1

Circulating miR-30b-5p is up regulated in Cavalier King Charles Spaniels affected by early myxomatous mitral valve disease

The authors thank the Editor and the Reviewers for their thoroughly second review of our study. 

We are very pleased about their appreciation of the work and their positive comments about the previous review. 

Thank you very much.

We have carefully considered all Reviewers’ comments and have tried to address them whenever we felt this was appropriate. 

Best regards

Review comments to the Author:

Reviewer #1: 

Thanks to the Authors for the work done in improving the manuscript, that can now be accepted for publication in this reviewer's opinion.

Thank you very much for your decision, we are very pleased.

Reviewer #2: I really appreciate the changes made by the Authors and the additional information provided. Such modifications increase the quality of the manuscript and the clinical relevance of its results. I thank the Authors for their great work. However, I have some residual comments that have been listed below. The majority of them are related to the way the sentences are structured and interconnected, especially in the Abstract (which is still a little bit complex to be read and may create some confusion to readers).

Thank you very much for your comments. We have changed the sentences as you suggested. 

Abstract

-lines 39: the word “abundance” I still present. As this term sounds a little be ambiguous, I would recommend to change it with another word, such as “amount” or a synonym.

Thank you, we have changed this term in all the text. 

-last 3 lines of the abstract: I would suggest to modify the end of the abstract (current version “Identifying dogs with early asymptomatic myxomatous mitral valve disease through the evaluation of miR-30b-5p as new clinical research data in this breed supports more focused screening programs”) in this way (or something similar): “Identifying dogs with early asymptomatic myxomatous mitral valve disease through the evaluation of miR-30b-5p represents an intriguing possibility that certainly merits further research. Studies enrolling a larger number of dogs with preclinical stages of MMVD are needed to expand further and validate conclusively the preliminary findings from this report”.

Thank you, we have changed this sentence as you suggested. 

Introduction

-Lines 67: I would suggest the word “interpreted” with the term “misinterpreted” (so that the sentence would become: “However, this test needs specialized equipment and well-trained operators to reduce interobserver variability, as valves affected by mild changes may be misinterpreted as normal”).

Thank you for this suggestion. We have changed the term. 

-Lines 77-85: I would suggest to change the sentences included in these lines in this way (or something similar): “Aberrant expression of miRNAs is associated with several human [18-21] and veterinary [22-25] disorders, including cancer and heart diseases. With specific regard to canine MMVD, the dysregulation of circulating miRNAs has been previously investigated by different approaches, including real-time quantitative PCR, microarray, and next-generation sequencing [26-31]. However, it should be noticed that most of the dogs enrolled in these studies were classified as American College of Veterinary Internal Medicine (ACVIM) stages C and D, while only one previous study has investigated circulating miRNAs in adult dogs with preclinical MMVD (i.e., ACVIM stages B1 and B2).”

Thank you for this suggestion. We have changed the sentence. 

-Line 89-93: I would suggest to change the sentences included in these lines in this way (or something similar): “It should be also highlighted that data from previous researches on dysregulation of circulating miRNAs in dogs with MMVD are partially biased by the limited sample sizes and the heterogeneity of the therapeutic protocols as well as study populations, as they were not specifically focused exclusively on one specific stage of the disease [28,29,31,33,34]”.

Thank you very much for this suggestion. We have changed the text as you suggested. 

-Lines 93-98: I would suggest to change the sentences included in these lines in this way (or something similar): “Given the above, this study was aimed at investigating the potential use of miRNA as biomarkers to identify dogs belonging to a specific ACVIM class, namely the stage B1. The decision to focus on this ACVIM class was driven by the fact that these dogs are most subjected to breed screening, and therefore are targeted as potential breeders. Moreover, the inclusion of a single ACVIM class offered the advantage of investigating miRNA in a very homogenous study population, which, in this case, is also free from the potential confounding effects of cardiovascular drugs (as ACVIM stage B1 do not need medical treatment for their underlying structural heart disease [reference of ACVIM guidelines 2019])”.

Thank you for this comment. We have modified this paragraph as suggested and we have added the correct reference. 

-Lines 98-104: I would suggest to deleted sentences included in these lines as the sound a little bit repetitive and not very useful in light of the topic and aim of the study (from “However, the disease goes undetected in subjects….” to “….will favour a focused screening of the subjects”).

Thank you, we have decided to follow your suggestion to delete this part from the introduction to be less repetitive. 

-Lines 106-110: I would suggest to modified these lines in this way (or something similar): “To further standardize our study population we also selected a specific canine breed, namely the CKCS, being the most commonly affected by MMVD [pertinent references]. Lastly, we focused our analysis on three specific miRNAs previously demonstrated to be associated with canine MMVD [29,31,33] and arbitrarily divided our study population in three distinct age categories (see below for further details) to explore how these miRNAs are modulated in the plasma of CKCS of different ages affected by MMVD at stage ACVIM B1”.

Thank you very much for this comment. We have modified this part of the introduction as you suggested. 

Discussion

-Lines 343-345: I have some perplexity about this sentence: “We demonstrated that miR-30b-5p could discriminate among ACVIM stage A CKCSs and ACVIM stage B1 CKCSs younger than 3 years, without audible heart murmurs and other clinical signs.” Specifically, my perplexity is about the “other clinical signs”. Indeed, dogs with MMVD at stage B1 do no have clinical signs at all, apart from the typical left-sided systolic heart murmur. Therefore, I would suggest deleting “and other clinical signs”.

Thank you, we have deleted “and other clinical signs”. 

-Lines 370-371: I would suggest to modify the sentence in this way: “without evading the echocardiographic evaluation, which undoubtfully remains the gold standard for MMVD diagnosis”.

Thank you, we have changed this sentence. 

-Line 375: correct the current version: you can write “contribute to treat” or “contribute treating”, while contribute to treating sounds wrong.

Thank you for this suggestion. We have changed it in “contribute to treat”.

-Line 376: replace the word “consequently” with the term “hopefully” or “theoretically”.

Thank you, replaced with “hopefully”. 

-Lines 377-378: Modify the sentence, so that the final result would be “If confirmed, this last result would be revolutionary as it would radically change the prognostic perspectives of…”

Thank you, we have modified the two sentences as you suggested. 

-Lines 385-386: I would further underline the fact that, currently, a widely available, rapid, reliable, not expensive kit for miRNAs analysis is lacking. Therefore, at the end of the sentence at line 386, I would write something similar to this: “Indeed, it should be note that, to date, a rapid, not expensive, widely available diagnostic kit to measure the miRNAs amount is not available in small animal practice; therefore, such an analysis is currently limited to few highly specialized laboratories”.

Thank you for this suggestion. We completely agree with the reviewer, and we have added this sentence. 

-Lines 406-410: I would suggest to change these sentences in this way (or something similar): “Furthermore, in human medicine it has been demonstrated a potential link between patient’s age and miRNAs profile. For example, it has been showed that miRNAs may play a role in the highly coordinated mechanisms regulating genes involved in children’s development [55]. Regrettably, in veterinary medicine, to date, such a link has not been investigated. For this reason, veterinary studies purposefully designed to take into account subjects’ age categories and investigate the link between age and miRNAs profile are needed”.

Thank you. We have modified this sentence as suggested. 

Conclusions

I believe that the conclusions of the study are overall out of topic. A study’s conclusion should primarily resume the key points relates to the main results of the study, so that readers can have some concise, clear take of message of the current research. In contrast, conclusions should not include speculations about the future and the possible, theorical application on the results of future studies which will be based on the present manuscript. Obviously, a sentence on the future prospective is desirable, but it should be a minimal part of the conclusion section, not its predominant/entire one. Therefore, I kindly invite Authors to rewrite their conclusions.

Thank you for this comment. We have modified the conclusion section as suggested trying to be more focused on the key points of this study:

“The growing interest in developing new molecular markers of heart disease in young dogs affected by MMVD has led to study the expression of 3 circulating microRNAs and their application as potential biomarkers in the plasma of CKCS with early asymptomatic MMVD (ACVIM stage B1). The hypothesis that healthy dogs have different microRNA expression profiles than B1 subjects of the same breed and that the microRNA profiles can differ within the same class among subjects of different ages have been confirmed. The amount of miR-30b-5p is significantly higher in ACVIM B1 CKCS than in ACVIM A subjects, and according to the age, the amount of miR-30b-5p is higher in group B1 younger than 3 years, B1 between 3 and 7 years, and B1 older than 7 years than in group A. These results lay the basis for future studies aimed at reaching more substantial data that will help the CKCS breeders in their targeted selection programs based on the echocardiographic evaluation and to obtain healthier subjects with a reasonable life expectancy. At the same time, highlighting the risk of developing the disease at an earlier stage will favour a focused screening of the subjects. To that end, identifying early biomarkers for premature MMVD would be a helpful addition”.

Thank you very much for your appreciation of the work, we are really pleased. Thank you for your precise and interesting comments. 

We believe that the work has been greatly improved by the reviews and we thank the reviewers for their indispensable work.

---

## [Decision Letter · Decision Letter 2]

27 Jun 2022

Circulating miR-30b-5p is up regulated in Cavalier King Charles Spaniels affected by early myxomatous mitral valve disease

PONE-D-22-07425R2

Dear Dr. Brambilla,

We’re pleased to inform you that your manuscript has been judged scientifically suitable for publication and will be formally accepted for publication once it meets all outstanding technical requirements.

Kind regards,

Vincenzo Lionetti, M.D., PhD

Academic Editor

PLOS ONE

Additional Editor Comments (optional):

Reviewers' comments:

Reviewer's Responses to Questions

**Comments to the Author**

1. If the authors have adequately addressed your comments raised in a previous round of review and you feel that this manuscript is now acceptable for publication, you may indicate that here to bypass the “Comments to the Author” section, enter your conflict of interest statement in the “Confidential to Editor” section, and submit your "Accept" recommendation.

Reviewer #1: All comments have been addressed

Reviewer #2: All comments have been addressed

2. Is the manuscript technically sound, and do the data support the conclusions?

Reviewer #1: Yes

Reviewer #2: Yes

3. Has the statistical analysis been performed appropriately and rigorously? 

Reviewer #1: Yes

Reviewer #2: Yes

4. Have the authors made all data underlying the findings in their manuscript fully available?

Reviewer #1: Yes

Reviewer #2: Yes

5. Is the manuscript presented in an intelligible fashion and written in standard English?

Reviewer #1: Yes

Reviewer #2: Yes

6. Review Comments to the Author

Reviewer #1: I believe the authors have done a great job improving the manuscript. I agree with the revised version of the manuscript.

Reviewer #2: I have really appreciated the great work made by Authors in order to match the Reviewers’ comments and suggestions, and I think that the manuscript can now be accepted for publication.

7. PLOS authors have the option to publish the peer review history of their article (what does this mean?). If published, this will include your full peer review and any attached files.

Reviewer #1: No

Reviewer #2: No

---

## [Editor Report · Acceptance letter]

30 Jun 2022

PONE-D-22-07425R2 

Circulating MiR-30b-5p is upregulated in Cavalier King Charles Spaniels affected by early myxomatous mitral valve disease 

Dear Dr. Brambilla:

I'm pleased to inform you that your manuscript has been deemed suitable for publication in PLOS ONE. Congratulations! Your manuscript is now with our production department. 

Kind regards, 

on behalf of

Prof. Vincenzo Lionetti 

Academic Editor

PLOS ONE